# Effect of Solvents, Stabilizers and the Concentration of Stabilizers on the Physical Properties of Poly(d,l-lactide-*co*-glycolide) Nanoparticles: Encapsulation, In Vitro Release of Indomethacin and Cytotoxicity against HepG2-Cell

**DOI:** 10.3390/pharmaceutics14040870

**Published:** 2022-04-15

**Authors:** Musaed Alkholief, Mohd Abul Kalam, Md Khalid Anwer, Aws Alshamsan

**Affiliations:** 1Nanobiotechnology Unit, Department of Pharmaceutics, College of Pharmacy, King Saud University, P.O. Box 2457, Riyadh 11451, Saudi Arabia; malkholief@ksu.edu.sa (M.A.); makalam@ksu.edu.sa (M.A.K.); 2Department of Pharmaceutics, College of Pharmacy, Prince Sattam Bin Abdulaziz University, Al-Kharj 11942, Saudi Arabia; m.anwer@psau.edu.sa

**Keywords:** PLGA, Indomethacin, nanoparticles, solvents, stabilizers, morphology, particle-size, encapsulation, drug release, cytotoxicity

## Abstract

A biocompatible, biodegradable and FDA-approved polymer [Poly lactic-*co*-glycolic acid (PLGA)] was used to prepare the nanoparticles (NPs) to observe the effect of solvents, stabilizers and their concentrations on the physical properties of the PLGA-NPs, following the encapsulation and in vitro release of Indomethacin (IND). PLGA-NPs were prepared by the single-emulsion solvent evaporation technique using dichloromethane (DCM)/chloroform as the organic phase with Polyvinyl-alcohol (PVA)/Polyvinylpyrrolidone (PVP) as stabilizers to encapsulate IND. The effects of different proportions of PVA/PVP with DCM/chloroform on the physiochemical properties (particle size, the polydispersity index, the zeta potential by Malvern Zetasizer and morphology by SEM) of the NPs were investigated. DSC was used to check the physical state, the possible complexation of PLGA with stabilizer(s) and the crystallinity of the encapsulated drug. Stabilizers at all concentrations produced spherical, regular-shaped, smooth-surfaced discrete NPs. Average size of 273.2–563.9 nm was obtained when PVA (stabilizer) with DCM, whereas it ranged from 317.6 to 588.1 nm with chloroform. The particle size was 273.2–563.9 nm when PVP was the stabilizer with DCM, while it was 381.4–466.6 nm with chloroform. The zeta potentials of PVA-stabilized NPs were low and negative (−0.62 mV) while they were comparatively higher and positive for PVP-stabilized NPs (+17.73 mV). Finally, drug-loaded optimal NPs were composed of PLGA (40 mg) and IND (4 mg) in 1 mL DCM/chloroform with PVA/PVP (1–3%), which resulted in sufficient encapsulation (54.94–74.86%) and drug loading (4.99–6.81%). No endothermic peak of PVA/PVP appeared in the optimized formulation, which indicated the amorphous state of IND in the core of the PLGA-NPs. The in vitro release study indicated a sustained release of IND (32.83–52.16%) from the PLGA-NPs till 72 h and primarily followed the Higuchi matrix release kinetics followed by Korsmeyer–Peppas models. The cell proliferation assay clearly established that the organic solvents used to prepare PLGA-NPs had evaporated. The PLGA-NPs did not show any particular toxicity in the HepG2 cells within the dose range of IND (250–500 µg/mL) and at an equivalent concentration of PLGA-NPs (3571.4–7142.7 µg/mL). The cytotoxicity of the hepatotoxic drug (IND) was reduced by its encapsulation into PLGA-NPs. The outcomes of this investigation could be implemented to prepare PLGA-NPs of acceptable properties for the encapsulation of low/high molecular weight drugs. It would be useful for further in vitro and in vivo applications to use this delivery system.

## 1. Introduction

Natural and synthetic polymers have been extensively used by researchers for drug delivery applications for decades [1,2,3,4,5]. Natural polymers (e.g., bovine serum albumin) have fallen out of favor due to impurity and cost effectiveness issues [6,7]. Synthetic polymers (e.g., poly d,l-Lactide-co-glycolic Acid, PLGA), on the other hand, have become more preferable to formulation scientists as drug delivery carriers, due to their superior biodegradability and biocompatibility properties [8,9]. A wide variety of synthetic polymers have been studied, with PLGA being one of the most used. PLGA is biocompatible and biodegradable, and has been explored in many drug delivery applications, such as controlled delivery carriers for peptide, proteins, antigens, genes, vaccines, growth factors and many other macromolecular therapeutics [8,10,11,12,13,14,15,16,17,18,19]. The frequent application of PLGA for drug delivery is due to its attractive mechanical and handling properties [20]. In addition, researchers prefer using PLGA because it has been approved by World Health Organization and Food Drug Administration (FDA) for human use since 1989 [5,7,9,21,22,23,24].

PLGA-NPs can be synthesized using different techniques, including emulsion solvent evaporation or diffusion, solvent displacement, salting-out, nanoprecipitation and many other techniques [8,25,26,27,28,29]. Of these methods, the single (*o*/*w*) or double (*w*/*o*/*w*) emulsion solvent evaporation are the oldest and most extensively used techniques to encapsulate the hydrophilic and hydrophobic drugs into the PLGA-based micro- or nanoparticles [6,9,22]. In this method, PLGA and Indomethacin (IND) are dissolved in organic solvent (chloroform/dichloromethane/ethyl acetate/acetone etc.) and then emulsified by the stabilizer present in the aqueous phase via homogenization followed by sonication [28]. The nano-droplets obtained are further treated by evaporating the organic solvent(s) before purifying and collecting the PLGA-NPs by washing with purified water (ultracentrifugation) for its further characterization [21,25]. The morphological and physical/physicochemical characteristics (particle size, its distribution, physical and stability, release profile of the encapsulated drug, etc.) of the NPs can be greatly influenced by the formulation parameters [30,31]. The above characteristics of PLGA-NPs might change by changing the type of organic solvent(s) used to solubilize the polymer and the type of emulsifier/stabilizer, and their respective concentrations. Other factors that play a role include the mechanical forces (such as homogenization/probe-sonication, magnetic stirring rate, etc.) used to emulsify the mixture of organic and solvents, and the duration for the complete evaporation of the organic solvent [27,30].

Indomethacin (IND) is a nonsteroidal anti-inflammatory drug (NSAID) used to relieve fever, pain, swelling and stiffness of joints in moderate to severe arthritic conditions such as rheumatoid, osteo and gouty arthritis or spondylitis. It is also used for other purposes including in the shoulder pain affected by tendinitis or bursitis. It acts by the reversible inhibition of cyclooxygenase enzymes (COX-I and COX-II), also known as prostaglandin-endoperoxide synthase. The COX-1 catalyzes the synthesis of prostaglandins and thromboxane, while COX-2 is articulated in response to inflammation and injury. Thus, the antipyretic, analgesic and anti-inflammatory actions of IND take place as a result of decreased prostaglandin synthesis. Unlike other NSAIDs, IND also inhibits the enzyme phospholipase-A2 (causes the release of arachidonic acid from phospholipids). The antipyretic effect of IND might be due to its action on the hypothalamus causing vasodilation, increased peripheral blood flow and thus the dissipation of heat [32]. The poor aqueous solubility of IND (0.937 mg/L at 25 °C) leads to poor bioavailability [33]. Thus, taking high doses of IND is needed for some conditions. High doses of this drug for prolonged use may cause intestinal bleeding and increase the risk of stroke or fatal heart attack or myocardial infarction [34]. The encapsulation of IND into nanoparticulate drug carriers is needed to improve its bioavailability and reduce its dose and dosing frequency.

In the present study, we investigated the effect of different formulation parameters on the physiochemical properties of the PLGA-NPs. We prepared four batches of PLGA-NPs using two different organic solvents, namely chloroform and dichloromethane (DCM), with two commonly used stabilizers, namely Polyvinyl alcohol (PVA) and Polyvinylpyrrolidone (PVP) and studied their influence on the particle size, its distribution and their morphological characteristics. We also investigated the effects of different concentrations and viscosities of the aqueous solution of the stabilizers on the PLGA-NPs. In addition, we utilized differential scanning calorimetry to check any polymer–stabilizer or drug interactions and the crystallinity of the encapsulated drug. To our knowledge, most of the literature has focused on the release behavior of the payload from the PLGA-NPs system. Thus, here we synthesized drug-loaded PLGA-NPs by the single emulsion solvent evaporation technique using IND as a model lipophilic drug. The physicochemical properties including the drug encapsulation, its loading and in vitro drug release and release kinetics were investigated in detail. Moreover, the toxicity of PLGA-NPs (with and without IND) was examined by performing the cytotoxicity using HepG2 cells.

## 2. Materials and Methods

### 2.1. Materials

Poly (d,l-lactic-*co*-glycolic acid) (50:50) with molecular weight 40,000–75,000 and Polyvinylpyrrolidone (PVP; Av. Mw ~ 55,000) were purchased from Sigma Aldrich Co. (St. Louis, MO, USA). Indomethacin (IND) was procured from Winlab, UK. Polyvinyl alcohol (PVA) with 17,200 Mw was purchased from AVONCHEM Ltd. (Macclesfield, Cheshire, UK). Dichloromethane (DCM) and chloroform were purchased from PANREAC QUIMICA SA, (Barcelona, Spain) and MERK (Darmstadt, Germany), respectively. Purified water was obtained through Milli-Q water purifier (Millipore, France) system.

### 2.2. Chromatographic Analysis of Indomethacin

The reverse phase (RP) HPLC-UV method was used for the analysis of IND at 241 nm wavelength [35,36,37]. “The HPLC system (Waters^®^, Model-1500-series controller, Milford, MA, USA) equipped with UV-detector (Waters^®^, Model-2489, dual absorbance detector, USA), binary pump (Waters^®^, Model-1525, USA), automated sampling system (Waters^®^, Model-2707 Autosampler, USA). The system was run by “Breeze software” (Breeze™ HPLC Systems From Waters Corporation, Milford, MA, USA). A C_18_ analytical column (Macherey-Nagel 150 × 4.6 mm, 5 μm) was used at room temperature. The chromatography was performed by isocratic elution of the mobile phase composed of 75:25, *v/v* of Acetonitrile and Milli-Q^®^ water (the pH of water was adjusted to 3.2 by Ortho-Phosphoric acid). The flow rate was 1 mL/min and the volume of injection was 30 μL. Total run time was 5 min and retention time (*R*_t_) for IND was 3.5 min. The calibration curve was constructed in the concentration range 0.1, 0.2, 0.4, 0.8, 1.6, 2.5, 5.0 and 10 µgmL^−1^. The straight-line equation obtained was *y* = 150,383*x* − 6548.7 with a correlation coefficient (*R*^2^) of 0.9991.

### 2.3. Preparation of PLGA Nanoparticles

The PLGA-NPs were prepared by the single emulsion solvent evaporation technique. Here, the PLGA-NPs were prepared using the oil-in-water (*o*/*w*) single emulsion solvent evaporation technique [38]. In brief, 40 mg of PLGA (50:50; Mol. Wt. of 40–75 kDa) and 4 mg of IND were dissolved into 1 mL of the organic phase consisting of either chloroform or dichloromethane (DCM). These organic phases were then emulsified after the addition of 5 mL of varying concentrations of (1%, 3%, 6% or 9%, *w/v*) of polyvinyl alcohol (PVA) or polyvinyl pyrrolidone (PVP) by pulse sonication using a probe sonicator at 40% power for 40 s (4 cycles, 10 s each) using a probe sonicator (Sonics & Materials, Inc., Newtown, PA, USA) on an ice bath. The organic solvent was evaporated by magnetic stirring at 800 rpm at room temperature for overnight to obtain the suspension of NPs. The suspended PLGA-NPs were purified after washing with Milli-Q^®^ water to remove the extra stabilizers and collected by ultracentrifugation at 15,000 rpm for 30 min at 4 °C (Preparative ultracentrifuge, WX-series by Hitachi Koki, Hitachinaka, Japan). Finally, the drug-loaded PLGA-NPs were also prepared by the same procedure, where 4 mg of IND was dissolved into the organic phase containing PLGA. The washing process was performed three times for the purification and collection of PLGA-NPs. The obtained NPs were then dispersed in 10 mL of Milli-Q water, freeze-dried and were stored at −80 °C [39] for further experiments.

### 2.4. Particle Size and Zeta Potential Measurement

The hydrodynamic diameter (mean particle size), size distribution and zeta potential of the developed PLGA-NPs were checked by Dynamic Light Scattering (DLS) measurement by Zetasizer Nano-Series (Nano-ZS, Malvern Instruments Limited, Worcestershire, UK) as per the reported methods [28,40]. Before starting the measurements, the suspensions of the NPs were suitably diluted with Milli-Q^®^ water (Millipore, France). For zeta potential, by considering the dielectric constant of the dispersant (i.e., 78.5 for water), the electrophoretic mobility was calculated at 25 °C by the software. The electrophoretic values were utilized to estimate the zeta potential by the software DTS V-4.1 (Malvern, Worcestershire, UK) installed in the system. All the measurements were performed in triplicate.

### 2.5. Morphological Characterization of PLGA-NPs

The shape and surface morphology of the PLGA-NPs was observed and determined by micrographs of the NPs obtained by Scanning Electron Microscopy (SEM) (Zeiss EVO LS10; Cambridge, UK) using the gold sputter technique. The freeze-dried PLGA-NPs were coated with gold using an ion-sputter (Q150R Sputter unit) from Quorum Technologies Ltd. (East Sussex, UK) in an argon atmosphere at 20 mA for 1 min. Observation and imaging was completed at 10 to 20 kV accelerating voltage and 8.5 mm of working distance. The magnification for the SEM images was kept at around 10 to 15 KX [28].

### 2.6. FTIR Spectral Analysis

Attenuated total reflectance (ATR) FTIR spectrum of solid samples (of pure drug (IND), PLGA 50:50, Empty PLGA-NPs and IND-loaded PLGA-NPs-5) were recorded using Bruker ALPHA spectrometer (Bruker Optics, Rosenheim, Germany). The spectrometer was equipped with a diamond prism with single bounce reflection ATR accessory. A thin layer of the solid sample was kept over the ATR accessory and the spectra were recorded from 4000–450 cm^−1^ wavenumber range using OPUS software (V-7.8), Bruker Optics, Rosenheim, Germany.

### 2.7. Viscosity Evaluation of the Surfactant Solutions

Viscosity study of the different concentrations of the stabilizers (PVA and PVP) was conducted to explore if there was a direct or indirect relationship between the viscosity and the characterization parameters [41]. The viscosity of the stabilizer solutions was evaluated by “Brookfield Viscometer (Brookfield Engineering Laboratories, Middleboro, MA, USA)” following as reported [42,43]. The viscosities (Pa.s) were determined at different shear rates (300–500 s^−1^ for PVA and 360–460 for PVP s^−1^) at 25 ± 1 °C temperature.

### 2.8. Encapsulation Efficiency and Drug-Loading Capacity

For the determination of the encapsulation efficiency (%EE) drug loading (%DL), 10 mg of drug-loaded PLGA-NPs were dispersed in 5 mL of Milli-Q water. The dispersion was centrifuged at 13,500 rpm at 10 °C for 15 min (PRISM-R, Labnet International Inc., Edison, NJ, USA). The supernatant was removed and the sediments of PLGA-NPs (precipitant) were washed by Milli-Q water in triplicate with 2 min of vortexing. Thereafter, 10 mL 1:1 (*v*/*v*) mixture of CHCl_3_ and DCM (to dissolve the PLGA and solubilize the IND) was added in the precipitant or pellet and vortexed. For complete extraction of the drug (Indomethacin, IND) from NPs, the mixture was subjected to ultrasonication (Model-3510, Branson Ultrasonic Corporation) for 20 min. The resulting suspension was further centrifuged for 10 min at 13,500 rpm. The supernatant was collected and the concentration of the drug was analyzed by the reported HPLC-UV method at 241 nm wavelength [35,36,37]. The %EE and %DL were calculated in triplicate according to the following equations (Equations (1) and (2)).
(1)$$\%\mathrm{EE}\;=\left(\;\frac{\mathrm{Amount}\;\mathrm{of}\;\mathrm{IND}\;\mathrm{in}\;\mathrm{NPs}\;\mathrm{precipitant}\;\left(\mathrm{mg}\right)}{\mathrm{Initial}\;\mathrm{amount}\;\mathrm{of}\;\mathrm{IND}\;\mathrm{added}\;\left(\mathrm{mg}\right)}\right)\times 100$$
(2)$$\%\mathrm{DL}\;=\left(\;\frac{\mathrm{Amount}\;\mathrm{of}\;\mathrm{IND}\;\mathrm{in}\;\mathrm{NPs}\;\mathrm{precipitant}\;\left(\mathrm{mg}\right)}{\mathrm{Total}\;\mathrm{mount}\;\mathrm{of}\;\mathrm{NPs}\;\left(\mathrm{mg}\right)}\right)\times 100$$

### 2.9. In Vitro Release and Kinetics Study

Based on the physicochemical features, two formulations from each batch were selected for the comparative in vitro release profile of IND from different PLGA-NPs. The in vitro release of IND from the selected and optimized PLGA-NPs was performed in in triplicate by the dialysis tubing method [44,45,46]. Phosphate buffer saline (PBS, pH 7.4) with 0.25% (*w*/*v*) of sodium lauryl sulphate (SLS) was used as the release media. An in vitro release study was also performed using dilute HCl solution (pH 1.2) with 0.25% (*w*/*v*) of SLS as release medium. A total of 10 mg of freeze-dried drug-loaded NPs was put in 1 mL of PBS mixed by vortexing and transferred into dialysis tubing. Both the ends of the dialysis tubing were tied and put into 100 mL capacity beakers containing 50 mL of the release medium. Beakers were placed in a shaking water bath (100 opm) where the temperature of water bath was maintained at 37 ± 0.5 °C. At stipulated time points 1 mL of the sample was taken out from each beaker and centrifuged at 13,500 rpm for 10 min. The supernatants was collected and 30 µL of each was injected into the HPLC-UV system for the analysis of released drug [37]. The cumulative amount drug released (%DR) was calculated by the following equation (Equation (3)), where DF stands for dilution factor.
(3)$$\%\mathrm{DR}=\frac{\mathrm{Conc}.\;\left(µg/\mathrm{mL}\right)\times \mathrm{DF}\;\times \mathrm{Volume}\;\mathrm{of}\;\mathrm{release}\;\mathrm{medium}\;\left(\mathrm{mL}\right)}{\mathrm{Initial}\;\mathrm{amount}\;\mathrm{of}\;\mathrm{IND}\;\mathrm{in}\;\mathrm{NPs}\;\left(\mu g\right)}\times 100$$

To check the kinetics and mechanism of IND release from the PLGA-NPs, the in vitro release data were fitted into different kinetic models [47,48]. The applied kinetic models include the zero-order (%DR versus time, Equation (4)), first-order (Log %D remaining versus time, Equation (5)), Higuchi matrix (%DR versus square root of time, Equation (6)), Korsmeyer–Peppas (Log fraction DR versus Log time, Equation (7)) and Hixson–Crowell (cube root of %D remaining in the polymeric matrix versus time, Equation (8)). From the values of slopes and co-efficient of correlation (*R*^2^) of kinetic plots obtained in different model equations, the release/diffusion-exponent (*n*-value) was calculated. The determination of *n*-values could provide an idea about the mechanism of drug release from the NPs [36,49,50].
(4)$$A_{t}=k_{0}t$$
(5)$$LogA_{0}-LogA_{t}=k_{1}t/2.303$$
(6)$$A_{t}=k_{HM}\sqrt{t}$$
(7)$$A_{t}=k_{KP}\;t^{n}$$
(8)$$\sqrt[{3}]{A_{0}}-\sqrt[{3}]{A_{t\;}}=k_{HC}t$$

In the above equations, *A_t_* is the amount of released IND at time *‘t’*, *A*_0_ is the initial amount of IND in the PLGA-NPs. The expression ‘*k’* is the rate constants (*k*_0_ for zero-order, *k*_1_ for first order, *k_HM_* for Higuchi matrix, *k_KP_* for Korsmeyer–Peppas, and *K_HC_* for the Hixson–Crowell model). The *n-*value in the Korsmeyer–Peppas equation is known as the diffusion or release exponent.

### 2.10. Differential Scanning Calorimetry (DSC)

The analysis of the physical state and possible complexation of the PLGA with the used stabilizers as well as the crystallinity of the encapsulated drug (IND) was checked through the differential scanning calorimetry technique through a DSC-8000 (Perkin Elmer Instruments, Shelton, WA, USA) at the heating rate of 10 °C per min. Around 4–5 mg of PLGA-NPs was weighed and put in aluminum pans and the lids were crimped. The pans were hermetically sealed and placed in the sample cell of the calibrated DSC instrument to check the melting temperature of the materials [40,51]. Both the reference and sample cells were continuously purged with N_2_ gas at the rate of 20 mL per min. The results were analyzed by the software Pyris™ (Version-11, PerkinElmer Inc., Houston, TX, USA) attached to the system. The thermal behavior of the materials used in the PLGA-NPs was examined at 10 °C per min of heating rate within 20 to 240 °C temperature range.

### 2.11. Stability of IND-Loaded PLGA-NPs

A short-term stability on drug-loaded freeze-dried PLGA-NPs (IND-PLGA-NPs-5) was performed [28,52,53]. The drug-loaded NPs (10 mg) were crammed into glass containers and stored at 4 ± 0.5 °C, 30 ± 1 °C (as per the Saudi Arabian climatic zone (IVa)) and 37 ± 1 °C for 1-month. The alteration in the particle size, polydispersity index, zeta potential, encapsulation efficiency (%EE) and drug-loading capacity (%DL) with storage time was determined at the 15th and 30th day to recognize the stability of the NPs. All the measurements were performed in triplicate.

### 2.12. In Vitro Cytotoxicity Studies

#### 2.12.1. Maintenance and Growth of Cell Line

The human hepatic cancer cell (HepG2) lines were cultured under CO_2_ (5%) at 37 °C using Dulbecco’s Modified Eagle’s Medium (DMEM), procured from UFC Biotech, Riyadh, Saudi Arabia). The media were accompanied with 1% mixture of “Penicillin-Streptomycin” (Thermo Fischer Scientific, Waltham, MA, USA), 10% Fetal Bovine Serum (FBS) and 1% L-Glutamine obtained from Alpha Chemika, Maharashtra, India and BioWest, Riverside, MO, USA, respectively, while the MTT was obtained from “Sigma Aldrich”, St. Louis, MO, USA.

#### 2.12.2. MTT Assay

The in vitro cytotoxic activities of the free drug (IND solution) and the IND-loaded PLGA-NPs were investigated as compared to the empty PLGA-NPs-5 using HepG2 cells by assessing their viability through an MTT assay [54]. The cell cultures in the phase of exponential growth were “Trypsinized” and diluted in DMEM to obtain a total cell count of 5  ×  10^5^ cells/mL. The suspension of cells were then transferred into flat-bottomed 96 well microplate in 100 μL medium and left overnight to attach. The medium was changed with 100 μL of fresh medium containing IND, IND-loaded NPs and empty PLGA-NPs-5 using the equivalent amount of NPs to obtain IND concentrations (5–1000 µg/mL) in the same media as the solvent. An equivalent amount of empty NPs was also used. The free IND was dissolved in DMSO and diluted with the media, keeping the concentration of DMSO not more than 1%, (*v*/*v*). Three wells were used as a negative control for cell viability, where only cells (without treatment) were suspended in the FBS free DMEM. Each product dilution was evaluated in triplicate. After 24 h, 48 h and 72 h of incubation, the cell suspensions were removed and the wells were washed with PBS. Then, 20 μL of MTT solution (at 5 mg/mL concentration in PBS) and 80 μL of the medium was added and incubated at 37 °C for 4 h. Subsequently, the culture fluid (containing media and MTT) was discarded leaving the precipitate of formazan crystals. The crystals were dissolved in 100 μL mixture (99.4 mL/0.6 mL/10 g) of DMSO, acetic acid and sodium lauryl sulfate at room temperature for 10–15 min. The microplates were observed and absorbance was measured at 570 nm using spectrophotometric microplate reader (Synergy HT, BioTek Inst., Winooski, VT, USA). The results were normalized as compared to the absorbance obtained for viable control cells (by considering as 100%) and expressed as plots of cells viability versus log concentration of IND. The IC_50_ values were determined for the employed cells against each product including the empty NPs using GraphPad Prism 5, San Diego, CA, USA.

### 2.13. Statistical Analysis

All the experiments were performed in triplicate (*n* = 3). The data were expressed as mean with standard deviation (SD). Data were analyzed and compared by the Paired *t*-test using GraphPad Prism 5, USA. The *p*-value, less than (*p* < 0.05) was considered as statistically significant.

## 3. Results and Discussion

### 3.1. Preparation of PLGA-NPs

For the preparation of NPs, we used PLGA as the main polymer mixed with either PVP or PVA, while either chloroform or DCM were used as the organic solvents. We employed the single emulsion solvent evaporation method which is helpful in optimizing the design of the NPs (size, morphology and surface properties) making them better candidates for drug delivery applications. In this method, the emulsification of the two phases and the stabilization (droplet protection) of the NPs are prime factors, which are greatly influenced by the type of stabilizers used and their quantity. PVA and PVP were chosen as stabilizers because they facilitate the formulation of smaller PLGA-NPs with a spherical shape and a narrow granulometric distribution even at higher concentrations, in addition to their ability to prevent the coalescence of particles to aggregate [55,56,57,58]. Although PVP-stabilized PLGA-NPs have shown similar features to PVA-stabilized NPs in terms of the encapsulation and loading of low molecular weight drugs [59], under similar conditions PVP-stabilized PLGA-NPs had higher efficiencies compared to PVA-stabilized NPs for the encapsulation and loading of proteins [59,60,61]. As for the type of organic solvents used, chloroform and DCM are the most commonly used to prepare the PLGA-NPs by the emulsification solvent evaporation method. However, reports have indicated that other solvents can be used, but mostly in combination, such as DCM-acetone that is used to dissolve rifampicin and DCM-acetone-ethanol used to dissolve Estradiol valerate [62]. Additionally, propylene carbonate has been utilized to reduce the toxic effect of benzyl alcohol to encapsulate estrogen into PLGA-NPs [57,58]. The solvent evaporation process during the preparation of PLGA-NPs is a critical step to encapsulate the drug(s) as it decreases the diffusion of drug(s) to the external aqueous phase, in addition to obtaining smaller particles [63,64]. The chloroform and DCM as organic solvents for PLGA are easily evaporated in a short duration even at a reduced pressure (vacuum rotational evaporator) or with magnetic stirring at normal atmospheric pressure [55,63]. The smaller particles are obtained due to the higher solvent-front kinetic energy that leads to the higher rate of diffusion of the organic phase at the interface and a higher rate of droplet dispersion, which results in smaller NPs [63]. Solvent evaporation causes a volume reduction in the dispersion phase which subsequently increases the density and viscosity of the dispersed NPs. This phenomenon prevents the coalescence and agglomeration of the NPs during the evaporation of organic solvents [64].

Using the proper preparation technique, such as single emulsion or double emulsion, is critical as the chosen method affects the physicochemical properties of NPs and the drug-loading capacity [65,66]. For example, the encapsulation and loading of lipophilic agents through the single emulsion technique supports the formation of smaller NPs than that of the double emulsion technique [17,67,68]. The employed emulsion solvent evaporation technique, also called the ultrasonication solvent evaporation technique, was first introduced by Vanderhoff et al. and has become a very popular method for the synthesis of polymer-based nanoparticles [7,62,65,69]. The process parameters in this technique, such as the volume of the aqueous phase containing stabilizer/emulsifier, magnetic stirring, homogenization speed and drug/PLGA ratio (in the case of drug-loaded NPs) can easily be controlled and optimized to obtain the NPs with an improved surface morphology and the desired size distribution [55,65]. This method has been used frequently to encapsulate low and high molecular weight drugs [27,28,70,71,72]. Moreover, smaller NPs could be obtained using a small quantity of PLGA and a large volume of a stabilizer-containing aqueous phase at a high homogenization speed while larger particles can be obtained at higher PLGA concentrations [55,56]. A high amount of polymer caused an increased viscosity of the internal phase, leading to its insufficient and non-homogenous dispersion into the aqueous phase during emulsification and resulting in larger NPs [55].

### 3.2. Effect of Formulation Factors on Particle Size and Zeta Potential

The PLGA-NPs prepared by the single emulsion solvent evaporation method produced particles with an average size in the range of 273–563 nm when PVA was used as the stabilizer and DCM was used as the organic solvent, whereas it was in the range of 317–588 nm when chloroform was used as the solvent. As for PVP, the mean particle size was in the range of 287–684 nm when DCM was used as the solvent, while it was in the range of 381–466 nm when chloroform was the solvent (Table 1).

Overall, the results of the particle size measurement indicate that PLGA-NPs prepared using PVA as stabilizer were smaller compared to PVP when DCM was used to solubilize the PLGA, with an exception at 1% PVP. This might be attributed to the good stabilizing property of PVP at this concentration (having optimum viscosity) with DCM. At the varying concentrations of PVA (1%, 3%, 6% and 9%, *w*/*v*) and using DCM as the organic solvent by keeping the other formulations factors constant (such as volume of solvents, stirring speed, amount of PLGA), an increase in the size of the NPs was found from 273–563 nm (Table 1). Similarly, an increase in particle size was observed from 317–588 nm as the concentration of PVA was increased and chloroform was used as the organic solvent. Contrary to this observation, the particle size was smallest (317 nm) at the highest (9%, *w*/*v*) concentration of PVA when chloroform was used, which might be due to the strong anchoring of the hydrophobic segment of PVA with the matrix of PLGA which remained closely associated with polymer surface [73]. The mechanistic effect of stabilizers on the size of PLGA-NPs might be due to the interpenetration of PVA/PVP and PLGA molecules during the formation of NPs [74,75]. These stabilizers might develop a uniform coating network throughout the surface of PLGA, where PVA cannot shield completely the negative surface charge (due to –COOH functional groups of PLGA) as compared to PVP. Thus, PVA caused a weak negative zeta potential [15], while PVP showed a relatively high positive zeta potential. In spite of the weak zeta potential, the PLGA-NPs were stabilized by the varying layers of PVA/PVP which surrounded the NPs by mechanism of steric hindrance [76].

The particle size and zeta potential distributions of the four NPs prepared with 1% stabilizers are represented in Figure 1. The values of polydispersity index and zeta potential are also summarized in Table 1. The zeta potentials of the NPs prepared with PVA showed low negatively charged surfaces, while most of the NPs prepared with PVP were found to have comparatively higher and positive surface charges. To check the effect of stabilizer on surface charges, the zeta potentials of 1% PVA and PVP solutions were also checked. The PVA solution showed slightly negative (−2.26 ± 0.49 mV) zeta potentials while it was positive (+5.06 ± 1.58 mV) for the PVP solution. Thus, we could state that the positive zeta potential of PVP solution might be the reason for the slightly higher zeta potentials of the PLGA-NPs developed using PVP as the stabilizer. The low polydispersity values in all cases of NPs (except in the cases where 6% and 9% PVP, which were 0.512 and 0.846, respectively) indicated the unimodal distribution of the particles in the dispersion medium.

As for the type of organic solvent, we observed that using DCM as the organic phase resulted in comparatively good-sized particles as compared to chloroform when NPs were prepared with the PVA at 1%, 3%, 6% and 9%, *w/v* concentrations. DCM produced uniform and smaller PLGA-NPs than chloroform (stabilized by PVA) which might be due to the comparatively low solubility/miscibility of DCM with water. On the other hand, when chloroform was used as an organic phase, it produced larger NPs. This contrasts with the previous work, where they reported that relatively smaller NPs were produced when they employed vitamin-E-TPGS as the stabilizer [22,27,40,77]. This can be explained by the degree of stabilization that occurs either from the stabilizer itself, or other formulation parameters (concentration of PLGA in organic phase, speed of magnetic stirring and homogenization and sonication power, etc.) [55,57,58]. While PLGA-NPs with larger particles sizes are generally unwanted, they still might have an advantage in cases where the high amount of lipophilic drugs (relative to the amount of PLGA) need to be encapsulated as compared to smaller particles (subject to the solubility of the stabilizer in aqueous phase and its miscibility with the organic phase). However, this should be pursued carefully as the release kinetics of the encapsulated drug(s) may be influenced [22,78,79]. Although both organic solvents have limited solubility in water, the higher solubility of chloroform in the aqueous phase as compared to DCM [80] facilitates its rapid diffusion from oil droplets towards the outer aqueous phase during the emulsification process, which causes the rapid precipitation of PLGA, consequently obstructing the formation of smaller sized NPs [15].

Overall, PLGA-NPs prepared by the single emulsion solvent evaporation technique produced smaller NPs with a smooth surface and reliable properties when the most commonly used stabilizer (PVA) was employed, even at higher concentrations with DCM as the organic solvent, which is in agreement with previous reports [81,82].

### 3.3. Effect of the Two Stabilizers on the Morphology of PLGA-NPs

The effect of PVA and PVP at different concentrations on the morphology of PLGA-NPs was observed in two different organic solvents, chloroform and DCM. An obvious effect in the morphological structure of the NPs was found, which can be observed in most of the SEM images as shown in the Figure 2 and Figure 3. At lower concentrations (1% and 3%, *w*/*v*) of PVA, NPs obtained had a spherical and regular shape with a smooth surface (Figure 2A,B, respectively) with no aggregation. This confirmed the relevance and suitability of the selected formulation parameters to prepare the PLGA-NPs. At higher concentrations (6% and 9%, *w*/*v*) of PVA, however, NPs tended to aggregate and form irregular shapes (Figure 2C,D, respectively). These aggregates might have been formed due to residual stabilizers, which in turn increase the viscosity of the aqueous phase, requiring extra washing of the NPs during ultracentrifugation [76]. Likewise, we observed a similar outcome when PVP was used as a stabilizer. At 1% and 3% *w/v* concentrations of PVP, we obtained NPs that had a smooth surface and spherical, regular shapes (Figure 3A,B, respectively). Whereas, the aggregated–irregular-shaped particles were found with higher concentrations (6% and 9%, *w*/*v*) of PVP (Figure 3C,D, respectively). On the other hand, when DCM was used as an organic solvent, the prepared PLGA-NPs with either PVA or PVP as stabilizer produced regular spherical-shaped NPs with smooth surfaces at all the four concentrations employed (Figure 4 and Figure 5). It is noteworthy that the population of NPs were very low but discrete and spherical at all concentrations of PVA with equal amounts of PLGA, which might be due to the strong emulsifying property of PVA even at low concentrations and its good surface stabilizing feature towards PLGA-NPs. The presence of PVA at the interface prevented the coalescence of nano-droplets and the aggregation of NPs, which might be due to reduction in the overall energy of the two phases by PVA [57,83]. This might be attributed to the high HLB value of PVA (>18) than that of PVP (14 to 16) [77,84] and might be due to the low viscosity of PVP as compared to PVA, even at the same concentrations of the two stabilizers. Overall, employing DCM as an organic solvent seems to produce more regular and spherical nanoparticles compared to chloroform, which is advantageous in terms of the shelf-stability of the PLGA-NPs due to the narrow, uniform and unimodal granulometric distribution of the particles [55,56,57,58]. Based on these results, NPs prepared with 1% of either PVA or PVP were used for further analysis.

### 3.4. Fourier Transform Infrared (FTIR) Analysis

The FTIR spectra were recorded to investigate the possible interaction (if any) between PLGA and IND during the encapsulation of IND into the PLGA-NPs. The FTIR spectra of PLGA, IND, empty PLGA-NPs and IND-PLGA-NPs-5 are presented in Figure 6.

The sample of pure IND demonstrated the characteristic bands at 1585 cm^−1^ (aromatic C=C stretching and 1680 cm^−1^ (carboxyl stretching). The C–Cl stretching at 831 cm^−1^, C-O stretching ether at 1219 cm^−1^ and 1066 cm^−1^. Low intensity peaks at 2123 cm^−1^ and 2230 cm^−1^ wavenumbers were possibly indicate the O–H (carboxy) stretching [85]. From pure PLGA the band at 1933 cm^−1^ was the characteristic peak of –C=O– stretching of aliphatic polyesters and also at 2030 cm^−1^ and 2124 cm^−1^ C=O bond stretching was observed. Moreover, the peaks at 727 cm^−1^ and 846 cm^−1^ were due to the C-H bending (vib). The IR scan of empty PLGA-NPs demonstrated characteristic bands at 1089 cm^−1^ and 1176 cm^−1^ which were due to the C-O stretching (ether) and peak at 1755 cm^−1^ was due to the aromatic C=C stretching. A less intense band appeared at 2023 cm^−1^ might be due to the –C=O– stretching of aliphatic polyesters (characteristic of PLGA) [86,87,88]. The IND-loaded PLGA-NPs has shown a prominent IR band at 2139 cm^−1^ (–C=O– stretching of aliphatic polyesters and a less prominent band at 2014 cm^−1^, are characteristic bands for PLGA). The characteristic bands of IND did not appear in the drug-loaded NPs-5, which might suggest the proper encapsulation of IND into core of the NPs. Moreover, from this IR spectral analysis, it was concluded that IND was compatible with the PLGA (main excipient) used to prepare the NPs. However, the broadening and less intense bands appeared in the IR spectrum of IND-PLGA-NPs-5, also suggesting the absence of a chemical interaction between the drug (IND) and the polymer (PLGA).

### 3.5. Viscosity-Shear Rate Relationship of Different Concentrations of PVA and PVP

In this study, the viscosity versus concentration relationship for PVA and PVP solutions at four different concentrations (1%, 3%, 6% and 9%, *w*/*v*) of each formulation, was investigated using Milli-Q water as the solvent [89,90]. An increment in the viscosity of aqueous solutions was noticed which was proportional to the concentration of the stabilizers (Figure 7). The concentration dependence on the viscosity was more pronounced in the case of PVA (Figure 7A) as compared to PVP (Figure 7B) at the same concentrations of both the stabilizer. Each solution showed pseudoplastic behavior, which was due to the concentration dependency of PVA and PVP.

The viscosity (Pa.s) of each solution was higher at the low shear rate (s^−1^) and the viscosity of aqueous solutions of PVA and PVP increased with increasing their concentrations, exceptionally in case of 3% PVA as compared to 1% PVA. This might be due to the fact that viscosity is highly time and temperature-dependent. The decreased viscosity of 3% PVA solution might be attributed to the unwanted delayed processing in viscosity determination, also the temperature of the solution might be increased [91]. Here, we observed that the viscosity of the PVA–aqueous solutions decreased less rapidly with increasing the shear rate compared to PVP solutions. This indicated that PVA solutions were more viscous than the PVP solutions at same concentrations. Although these are well known phenomena, in this study we tried to correlate the effect of viscosity of the stabilizers on the size and morphology of PLGA-NPs. The measurements of viscosity of polymer(s) manifests the molecular interactions between the polymer (here, PVA and PVP) and the solvents, and from this the degree or magnitude of the interaction could be predicted [41,92]. The relatively smaller sized PLGA-NPs were obtained at lower concentrations of PVA and PVP which may have resulted from the low viscous solution of the stabilizers. Due to the low viscosity of the aqueous phase, their good interaction with the PLGA in organic phase led to the stress-free formation of PLGA-NPs during the organic solvent evaporation [15]. Viscosity of PVP aqueous solutions exhibited Newtonian flow at low shear-rate and lower concentrations, and shear-thinning at high shear-rate and higher concentrations (Figure 7B). The findings related to the effect of shear rate on the viscosity of aqueous solutions of stabilizer could be useful in understanding their effect in the development of PLGA-NPs [93,94].

### 3.6. Loading of Drug (IND) and In Vitro Release

Considering the facts of the above findings about the empty PLGA-NPs, a few drug- loaded formulations were developed to encapsulate a model lipophilic drug (indomethacin, IND). Where, PVA/PVP (1% and 3% *w*/*v*), PLGA (40 mg) and IND (4 mg), DCM and CHCl_3_ were used as organic solvent(s) to dissolve the polymer and the drug. The developed formulations were further characterized.

#### 3.6.1. Encapsulation Efficiency and Drug Loading Capacity

The encapsulation efficiencies and drug loading capacities of the selected PLGA-NPs of the four batches (Batch-1 to Batch-4) are summarized in Table 2. The physicochemical properties indicated the acceptable size of PLGA-NPs, good polydispersity with high encapsulation of IND at 1% PVA (stabilizer) and DCM as organic solvent. The mean size of PLGA-NPs-5, polydispersity and zeta-potential were 275.4 ± 8.5 nm, −0.157 ± 0.044 and −1.13 ± 0.29 mV respectively, with 72.96 ± 4.59% encapsulation, 6.63 ± 0.41% drug loading and highest amount (52.16 ± 3.83%) of cumulative drug release at 72 h as compared to other formulations. Although the highest encapsulation and drug loading (74.86 ± 4.78% and 6.81 ± 0.45%) was found with PLGA-NPs-6 (3% PVA and DCM) but the cumulative release was poor (37.88 ± 2.11%). Similarly, the size of PLGA-NPs-13 (286.9 ± 13.2 nm) was comparable to PLGA-NPs-5 and slightly high positive surface charge (10.4 ± 3.94 mV) but other parameters including the drug release were not sufficiently good. Overall, at similar preparation conditions the PLGA-NPs-5 showed satisfactory physiochemical characteristics; thus, it was assumed as the best one amongst the eight optimized PLGA-NPs of the four batches.

#### 3.6.2. In Vitro Drug Release and Release Kinetics

A sustained release of IND from all the optimized PLGA-NPs was found at pH 7.4, as shown in Figure 8). The in vitro release profiles of IND from the selected formulations of the four batches (PLGA-NPs-1 with 1% PVA and PLGA-NPs-2 with 3% PVA from Batch-1, PLGA-NPs-5 with 1% PVA and PLGA-NPs-6 with 3% PVA (Figure 8A) from Batch-2, PLGA-NPs-9 with 1% PVP and PLGA-NPs-10 with 3% PVP from Batch-3 and PLGA-NPs-13 with 1% PVP and PLGA-NPs-14 with 3% PVP from Batch-4 (Figure 8B) were summarized in Table 2. For the comparative release assessment, two time points, i.e., 12 h and 72 h were chosen for all the formulations. The cumulative release of the drug from PVA group PLGA-NPs at 12 h were almost same without any significant difference (*p* < 0.05), a highest amount 52.16 ± 3.83% of drug was released from PLGA-NPs-5 (1% PVA), which was significantly (*p* < 0.05) higher from the rest of the three formulations of PVA group. The use of DCM as an organic solvent to prepare this formulation provided a good encapsulation of IND which might have resulted in its higher and sustained release in PBS with 0.25% SLS.

At 12 h, the release of IND from PVP group formulations prepared with CHCl_3_ were almost same (15–16%) with no significant difference (*p* < 0.05), while a slightly improved release (19.4%) was found with the formulations prepared with DCM. However, the highest amount 51.6% of drug was released from PLGA-NPs-14 (3% PVP), which was slightly higher from the rest of the three formulations of PVP group. The higher concentration of PVP (3%, *w*/*v*), provided better stabilization to this formulation, which might be the reason for improved and sustained release of IND in the selected release medium.

The in vitro release profiles of IND in acidic conditions (pH 1.2) from the selected PVA group and PVP group formulations of the four batches are illustrated in Figure 9A, B, respectively, while the cumulative amount of the drug released at 12 h and 72 h is also summarized in Table 2. The highest amount (42.16%) of drug was released at 72 h from PLGA-NPs-1 (1% PVA with CHCl_3_), slightly less amount (40.06%) was released from PLGA-NPs-5 (1% PVA with DCM) and 39.2% was from PLGA-NPs-6 (3%PVA with DCM) which were significantly (*p* < 0.05) higher from the PLGA-NPs-2 (3%PVA with CHCl_3_) of PVA group formulations. Similarly, for PVP group formulations, the highest amount (43.16%) of drug was released at 72 h from PLGA-NPs-41 (3% PVP with DCM), slightly less (38.14%) was released from PLGA-NPs-9 (1% PVP with CHCl_3_) and an almost equivalent amount (32.5% and 35.81%) was released from PLGA-NPs-10 and PLGA-NPs-13, respectively.

Overall, the cumulative amount of drug released from most of the PVA and PVP group PLGA-NPs in acidic release medium followed a biphasic release pattern as illustrated in Figure 9A,B. Initially, a burst release was found from all the formulations followed by a sustained release of the drug, while a sustained release pattern was observed throughout the experiment performed at pH 7.4. Around 21.3–33.9% and 21.4–28.01% of drugs were released within 6 h from PVA and PVP group formulations, respectively. At 12 h, 30.38–41.27% and 26.46–39.89% drugs were release from PVA and PVP group formulations, respectively, while a maximum 42.01% (from PVA group) and 42.61% (from PVP group) were released in 48 h. After 12 h, clearly the prolonged and sustained drug release occurred till 72 h. An initial burst release might be beneficial for the use of IND in terms of acute inflammatory conditions, as it would assist the quick achievement of the therapeutic level, followed by a sustained release to maintain the therapeutic concentration of IND. The initial burst release of a drug might be due to loosely bound or surface-adsorbed drugs and the fast dissolution of IND in an acidic environment (pH 1.2). Secondly the sustained release could be due to the diffusion of the drug across the PLGA matrix, following the drug dissolution and polymer erosion.

The release of the drug at 12 h was very high at an acidic pH as compared to pH 7.4 as shown in Table 2. The release of the drug at acidic and basic conditions are almost equal from PLGA-NPs-1, PLGA-NPs-2 and PLGA-NPs-6 at 72 h study. While from other formulations the drug release was comparatively higher at pH 7.4 at 72 h, which could be attributed to high drug release during the initial hours at acidic condition, might saturated the release even after maintaining the sink condition. Thus, further release from the formulation could not have occurred, so the release in acidic condition was low in some formulations. Moreover, the degradation mechanism of PLGA at pH 7.4 takes place “inside-out” and at acidic pH it occurs as “outside-in”. At pH 7.4, in some PLGA-NPs the pore formation and surface pitting might occurred. Whereas, at an acidic pH the PLGA-NPs maintained their smooth surfaces during the process of polymer degradation and rupturing. The rupturing of the PLGA-NPs was accredited to the crystallization of the oligomeric degradation of PLGA due to its low solubility at acidic pH. Moreover, Zolnik and Burgess [95], reported that the degradation of PLGA taken place in an extra homogeneous pattern at acidic pH as compared to pH 7.4. This might be the result of entire NPs experienced a close-to-uniform pH at acidic condition. However, at pH 7.4, the local micro environmental pH within the PLGA-NPs was varied significantly due to accumulation of acid oligomers. Thus, a heterogeneous degradation resulted in the random creation of pores within PLGA-NPs degraded at pH 7.4 which was not occurred during their degradation acidic pH.

The sustained release of IND from the PLGA-NPs was obtained from initial to 72 h, when PBS (pH 7.4) with 0.25% (*v*/*v*) SLS was used as release medium. Thus, only the release data obtained from this experiment was subjected to different kinetic models to check the mechanism of drug release from PLGA-NPs. The plots obtained in the applied models were represented in the Appendix A. The calculated values of kinetic parameters including the rate constants of different reactions (*k*_0_ for zero-order, *k*_1_ for first order, *k_HM_* for Higuchi-Matrix, *k_KP_* for Korsmeyer–Peppas and *K_HC_* for Hixson-Crowell models) were summarized in Table 3. Based on the best fit model with highest correlation coefficient (*R*^2^) value, it was found that the selected NPs from the four batches primarily followed the Higuchi matrix release model, indicating the diffusion-controlled drug release from the PLGA-NPs, which was further substantiated by the mathematical calculation of the release-exponents (Table 3). The Higuchi-Matrix mathematical model in this investigation recommended that the initial drug concentration in the PLGA-NPs was much higher as compared to the solubilization of PLGA, even after uninterrupted maintenance of the sink condition throughout the experiment. Moreover, the diffusion of drug (IND) remained persistent and the swelling of polymer-matrix of the NPs was not significant, which was also suggested in some previous reports [49,96].

Apart from the Higuchi-Matrix model, the second-best fit model was the Korsmeyer–Peppas with second highest *R*^2^-values (Table 3). Where, linearity in the release curves of IND till 72 h (Log fraction drug released versus log time) with the higher *R*^2^ values were found (Appendix A). This also indicated the sustained release feature of the PLGA-NPs. The diffusion-exponent values were fallen between 0 and 0.5 for all the formulations of four batches (Table 3), indicated a Fickian diffusion mechanism of IND release. Among the applied release kinetic models, for all the PLGA-NPs from the four batches, the Higuchi-matrix model was appeared as the best fitted one followed by Korsmeyer–Peppas model. This model described the progressive loss of PLGA-NPs with time which was due to polymer erosion, and the release of IND from PLGA-NPs was occurred due to the diffusion of release medium into the matrix of PLGA. Concurrently, swelling and erosion of polymer matrix occurred, which might be the reason of sustained release of the encapsulated IND in the present investigation, which was also reported in previous report [97].

For the preferred PLGA-NPs-5, the *R*^2^ and *n* values were 0.999 and 0.0282 (Higuchi matrix model); 0.994 and 0.2563 (Korsmeyer–Peppas model) as shown in Table 3. In both the model, though the *n*-values were lower than 0.5, yet, these values still indicated the diffusion-controlled release mechanism for IND from the PLGA-NPs [49,98]. As, the *n*-values were <0.45 in the present case of IND-loaded spherical PLGA-NPs, a Fickian-diffusion release mechanism was directed [97,99]. The non-linear curve fitting of in vitro release data obtained for PLGA-NPs-5, using the Higuchi matrix model (primarily) and Korsmeyer–Peppas model (secondary), shows that the release constant (*k*, indicating the rate of drug release) was directly proportional to the diffusion exponent. Therefore, the drug release was depended on the structural and physical properties of the IND and the PLGA-matrix. The highest values of “*k*” (*k_HP_*, 2.08 × 10^−1^ and *k*_HM,_ 5.26 × 10^−2^ h^−1^) were found with PLGA-NPs-5 (Table 3). These findings were in agreement with the previous reports for the release of vancomycin from chitosan-alginate microparticles [100] and from the microporous calcium phosphate ceramics [101].

### 3.7. DSC Analysis

Differential Scanning Calorimetry was conducted on the optimized lyophilized NPs (PLGA-NPs-5) to determine any possible complexation of the polymer (PLGA) with the stabilizers (PVA and PVP), and to check the crystallinity of the encapsulated model drug (IND). The overlay scans of DSC were represented in Figure 10. A sharp endothermic peak at 157 °C for IND was appeared as shown in Figure 10A, indicates the crystalline characteristic of the drug. The reported melting point of indomethacin is 158 °C, a slight variation in the melting temperature indicated the purity of the drug. The endothermic peaks of PLGA (Figure 10B), PVA (Figure 10C) and PVP (Figure 10D) appeared at 60 °C, 202 °C and 152 °C respectively. The peak of IND was almost diminished in the PLGA-NPs suggesting that the drug was encapsulated into the core of the NPs. The appearance of a very low intensity melting peak (at around 158 °C) in case of PLGA-NPs-5 (Figure 10E), might be due to the presence of surface adsorbed drug in the crystalline state.

Although, no endothermic peak of stabilizer was noticed in the DSC scan of the lyophilized PLGA-NPs, indicating that the stabilizer was completely washed out during the purification steps by ultracentrifugation or the traces of the stabilizer (if remained), was not in the crystalline state. Furthermore, non-appearance of any new peaks throughout the scanning of the NPs, indicating the good compatibility among all the excipients used to develop the NPs.

### 3.8. Stability Studies

The results of the stability study (Table 4) indicate that the freeze-dried PLGA-NPs-5 were stable at different temperatures for 1 month. Almost no changes in the mentioned characteristic parameters for the PLGA-NPs were found stored at 4 °C. A slight variation in particle sizes were noted in the samples store at 37 °C, and such changes were non-significant. No obvious change in the particle size was associated with the stabilizer’s protection effect on the PLGA-NPs shell, which was also reported in the previous short term stability on drug loaded PLGA-NPs [28,52].

### 3.9. Cytotoxicity Study by MTT Assay

Drug encapsulation into PLGA-NPs provides a good solution to minimize the potential toxicity while retaining the therapeutic efficacy of numerous loaded drugs including chemotherapeutics, analgesics and NSAIDs [102]. Here, we sought to encapsulate IND into PLGA-NPs to reduce the toxicities associated with its free delivery and due to the organic solvents used in its preparation, by evaluating the cell survival rate during MTT assay. The cytotoxicity of IND, empty PLGA-NPs and IND-loaded NPs as well as PLGA-only were evaluated after an incubation with the HepG2 cells for 24 h, 48 h and 72 h. The results for IC_50_ were represented in Figure 11, which indicated a decrease in the cell viabilities with increasing concentrations of the tests.

The histograms of the percentage cell viability against varying concentrations of IND and the NPs were represented in Figure 12. As shown in Figure 12, cell proliferation of IND was examined at 5–1000 µg/mL concentrations, where the observed IC_50_ were 242.4, 129.9 and 80.06 µg/mL at 24 h, 48 h and 72 h, respectively. A report of Xiong et al., 2013, indicated that the PLGA-NPs was not significantly toxic up to 300 μg/mL against RAW264.7 and BEAS-2B cells [103]. Likewise, in the present investigation, we found that the IND-loaded PLGA-NPs-5, has shown increased cell viability at an equivalent drug concentration (5–1000 µg/mL) and the observed IC_50_ were 430.5, 372.2 and 260.9 µg/mL at 24 h, 48 h and 72 h, respectively. Moreover, the cell proliferation assay conducted using empty-PLGA-NPs at equivalent concentration of IND (i.e., the amount of empty-PLGA-NPs those containing the same amount of IND in the drug-loaded NPs) has shown IC_50_ values of 1096, 605.2 and 410.9 µg/mL at 24 h, 48 h and 72 h, respectively. Thus, even at these high concentrations of PLGA, the cells were viable at IC_50_ of 410.9 µg/mL when the duration of treatment was prolonged (72 h). This might be due to the time-dependent effect of PLGA-NPs on the cytotoxicity.

The MTT assay results clearly demonstrating the reduction in toxicity of IND by 66% in case of formulation (IND-PLGANPs) and empty PLGA-NPs with IC_50_ of 7142.7 µg/mL. Thus, we can assume that even at this very high concentration of PLGA (empty PLGA-NPs/IND-PLGA-NPs-5), the product was non-toxic as even at these high concentrations of PLGA, the cells were viable and the IC_50_ was significantly reduced as the duration of treatment was prolonged (i.e., 410.9 µg/mL at 72 h), indicating the time-dependent effect of PLGA-NPs on the cytotoxicity.

At high concentrations the PLGA-NPs was toxic, this might be associated with the physicochemical properties including the particle size, zeta-potential and selective targeting of the PLGA-NPs. These play an important role in terms of its efficacy in biological systems. In previous reports, it was found that the increasing the particle sizes, the IC_50_ of cytotoxicity was also increased, which suggests that small sized NPs encouraged higher cytotoxicity at low IC_50_ values [104,105]. The cytotoxicity results show that PLGA-NPs induced an anti-proliferative effect at an average particle size of 189 nm, while in the present investigation the size of the tested NPs was 275.4 nm, this might be one reason for its toxic nature at some concentrations. Even though our formulations did not exhibit any potential toxicity of the PLGA-NPs against HepG2 cells, might be due to its non-targeting and high zeta-potential properties [103,104,105].

The HepG2 are liver-derived human cell lines those display abundant parenchymal cell functions containing the metabolism of polycyclic aromatic hydrocarbons. The observed proliferation rate of HepG2 (i.e., doubling time) was measured, and it was found that these cells were doubled after 48.95 h at 5 × 10^5^ cells/well, cell density of HepG2 cells. The highest live cells were observed at 72 h, and these results were similar to Miura et al., 1999, who seeded 4 × 10^5^ cells/well in 6-well plates, and they believed that this was a more appropriate cell number for 6-well plates [106]. The doubling time of HepG2 cells was 48.95 h in the present investigation, which was consistent with other studies, where the doubling time of HepG2 was around 2.05 days (49.2 h) [107,108,109]. The growth rate at 24 h, 48 h, and 72 h were 53%, 52%, and 49%, respectively, and there was no major change in the growth rate with cell viability of more than 94.27% at 72 h. Thereby, no significant change in doubling time was found.

Overall, the PLGA-NPs in the present investigation did not show any particular toxicity at most of the concentrations and could not influence the normal growth of HepG2 cells within the IND dose range (250–500 µg/mL) and at equivalent amount of PLGA-NPs (3571.4–7142.7 µg/mL). Conclusively, an ideal size, zeta-potential, dose, duration of treatment and targeting ability of PLGA-NPs to deliver the encapsulated drugs at desired site are essential to develop an efficient and safe drug carrier.

A significantly (*p* < 0.05) high IC_50_ value for the PLGA-NPs indicated that the FDA-approved polymer used to prepare the NPs was non-toxic. The above findings also indicated that the complete evaporation of the organic solvent (DCM) used to prepare the NPs. Moreoverthe cytotoxicity of hepatotoxic drug (IND) can be reduced by its encapsulation into PLGA-NPs, which was in agreement with previous reports [102,110].

## 4. Conclusions

We have acknowledged some process variables that are decisive to obtain proper sized, uniformly distributed PLGA-NPs with a smooth surface and a solid dense structure discrete. This study has shed light on the influence of preparation condition on physicochemical properties of PLGA-NPs, following the single-emulsion solvent-evaporation method to prepare the NPs. The PVA at lower concentrations was proved to be a good stabilizer to obtain an optimized PLGA-NPs with excellent size and surface properties, when DCM was used as organic solvent to dissolve the polymer. Controlling the particle size and morphology of the NPs may have a potential impact in tailoring the drug delivery applications of such nano-carriers. The developed and characterized PLGA-based NPs as controlled delivery system in the present investigation was stable and would have good potential applications for the loading of hydrophilic and poorly soluble lipophilic therapeutic agents. A short-term storage physical stability indicated that the optimized PLGA-NPs was stable at different temperatures for 30 days. A prolonged release of the IND from the optimized NPs suggested the sustained and controlled release property of PLGA. A significantly (*p* < 0.05) high IC_50_ value for the NPs during the cytotoxicity study, indicated that the developed PLGA-NPs was non-toxic. Moreover, the adverse effect associated with the cytotoxic drugs such as IND (a hepatotoxic drug) could be overcome by their encapsulation into the PLGA-NPs. Nonetheless, further studies are needed to substantiate the current outcomes, which we are currently conducting in our lab that may further extend the utility of PLGA-NPs.

## Figures and Tables

**Figure 1 pharmaceutics-14-00870-f001:**
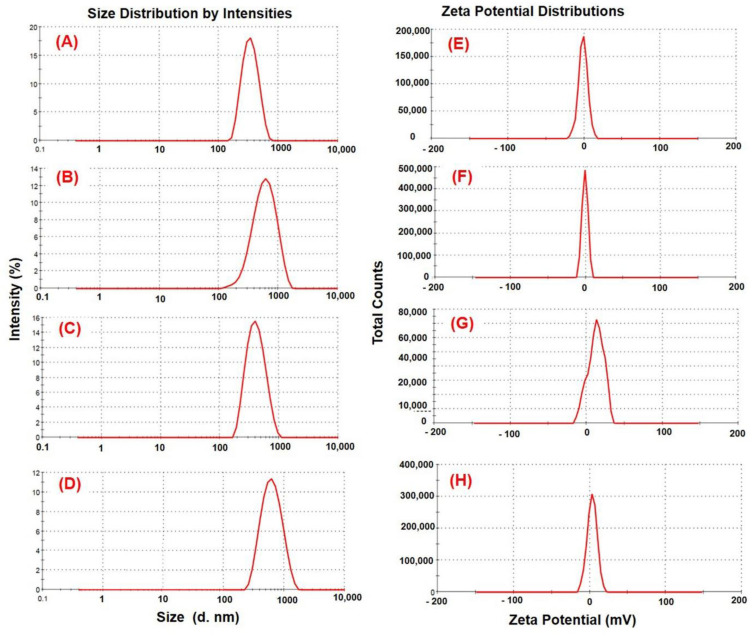
Particle size and zeta potential distribution curves of PLGA-NPs prepared with: (**A**) 1% PVA and chloroform; (**B**) 1% PVA and DCM (**C**) 1% PVP with chloroform (**D**) 1% PVP with DCM and (**E**–**H**) are their zeta potentials, respectively.

**Figure 2 pharmaceutics-14-00870-f002:**
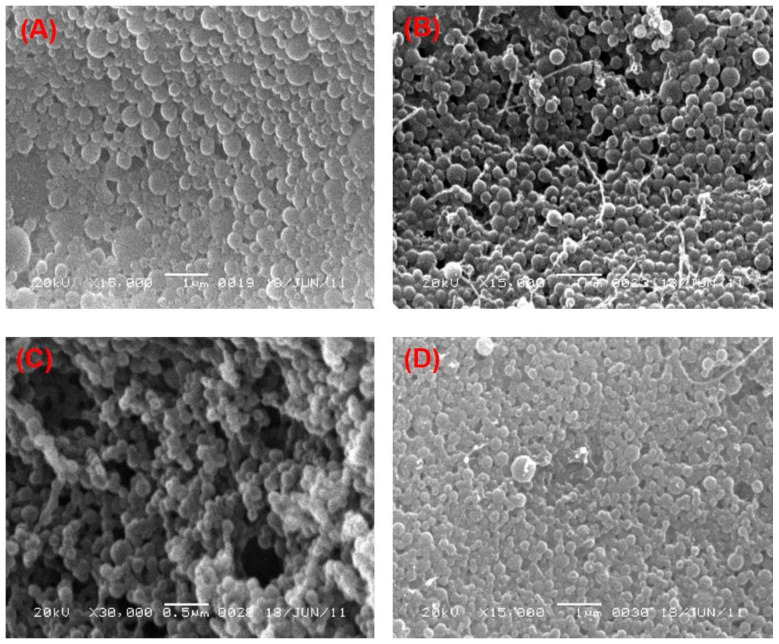
SEM images of the PLGA-NPs prepared with chloroform as organic solvent with different concentrations of PVA in the aqueous phase. (**A**) 1%, *w/v* PVA; (**B**) 3%, *w/v* PVA; (**C**) 6%, *w/v* PVA and (**D**) 9%, *w/v* PVA.

**Figure 3 pharmaceutics-14-00870-f003:**
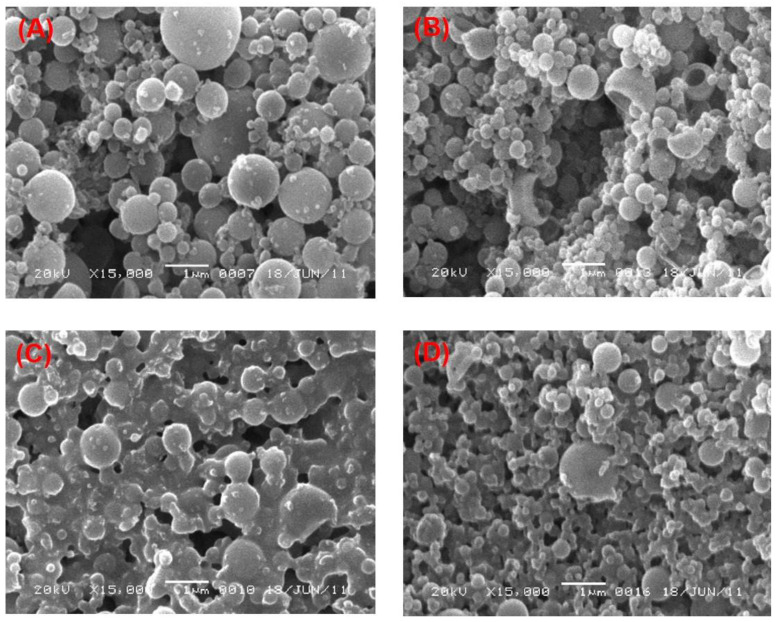
SEM images of the PLGA-NPs prepared with chloroform as organic solvent with different concentrations of PVP in the aqueous phase. (**A**) 1%, *w/v* PVP; (**B**) 3%, *w/v* PVP; (**C**) 6%, *w/v* PVP and (**D**) 9%, *w/v* PVP.

**Figure 4 pharmaceutics-14-00870-f004:**
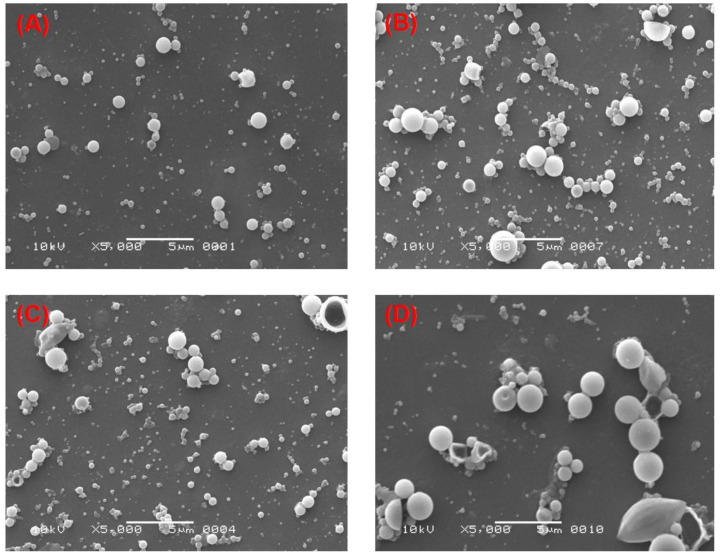
SEM images of the PLGA-NPs prepared with DCM as organic solvent with different concentrations of PVA in the aqueous phase. (**A**) 1%, *w/v* PVA; (**B**) 3%, *w/v* PVA; (**C**) 6%, *w/v* PVA and (**D**) 9%, *w/v* PVA.

**Figure 5 pharmaceutics-14-00870-f005:**
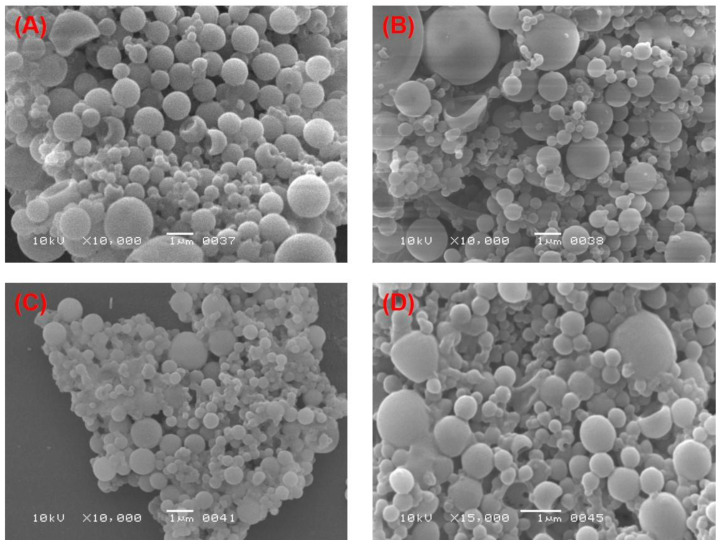
SEM images of the PLGA-NPs prepared with DCM as organic solvent with different concentrations of PVP in the aqueous phase. (**A**) 1%, *w/v* PVP; (**B**) 3%, *w/v* PVP; (**C**) 6%, *w/v* PVP and (**D**) 9%, *w/v* PVP.

**Figure 6 pharmaceutics-14-00870-f006:**
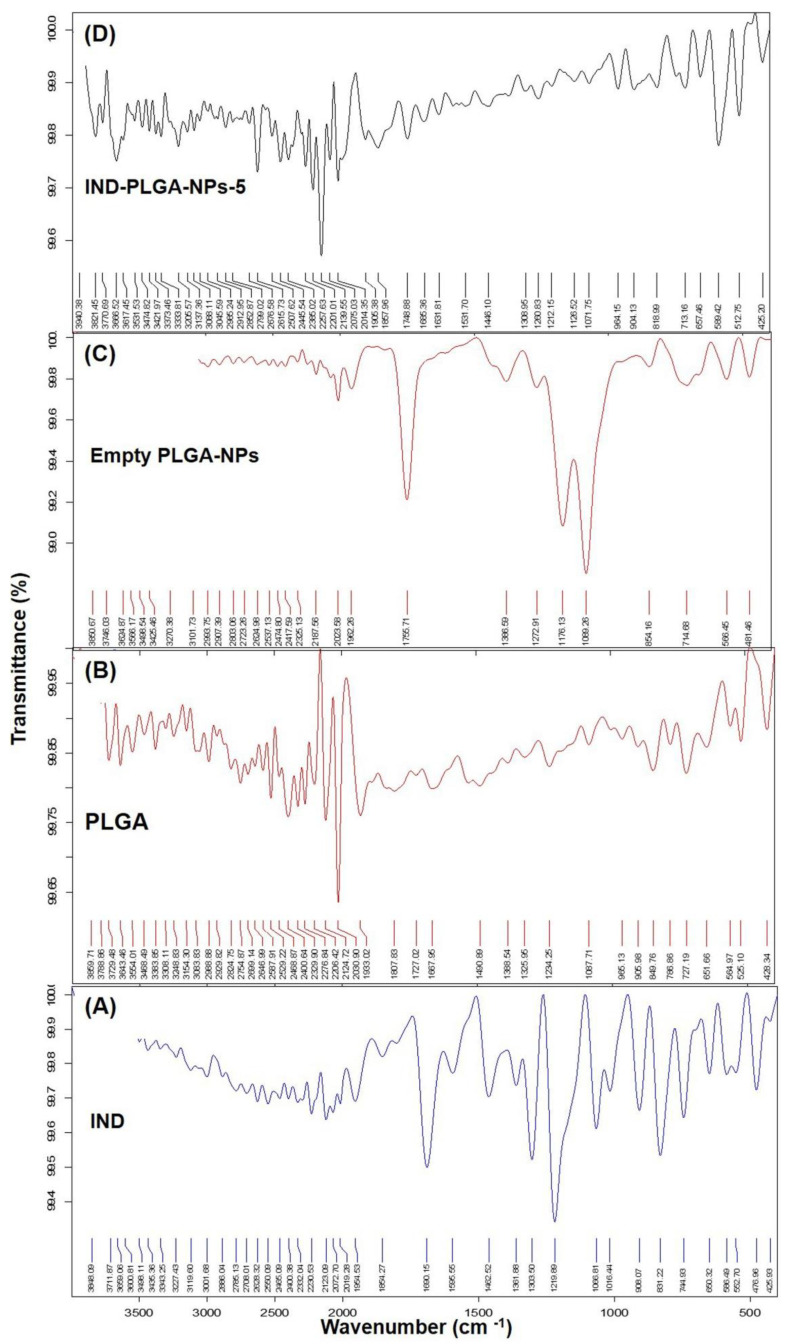
FT-IR spectra of pure IND (**A**), PLGA (**B**), empty PLGA-NPs (**C**) and IND-loaded PLGA-NPs-5 (**D**).

**Figure 7 pharmaceutics-14-00870-f007:**
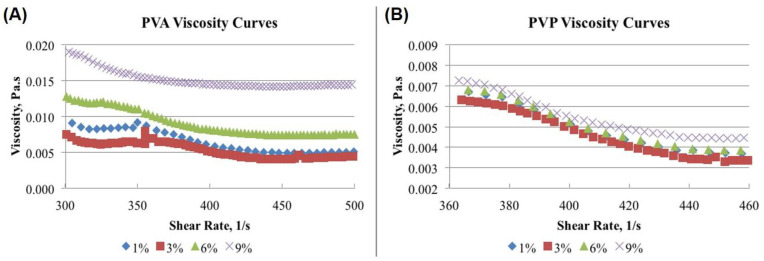
Relationship between viscosity and shear rate in the aqueous solutions of PVA (**A**) and PVP (**B**).

**Figure 8 pharmaceutics-14-00870-f008:**
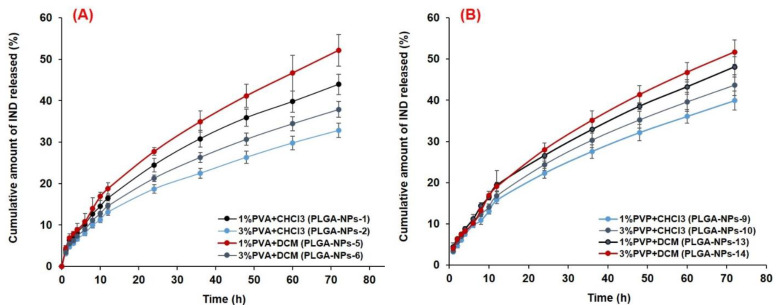
In vitro release profiles of IND from different PLGA-NPs in PBS (pH 7.4). Prepared with different concentrations of stabilizers, (**A**) PVA and (**B**) PVP.

**Figure 9 pharmaceutics-14-00870-f009:**
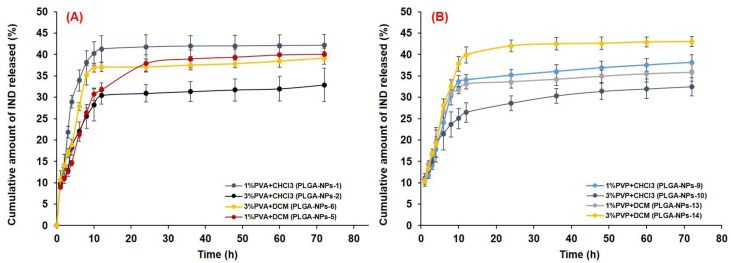
In vitro release profiles of IND from different PLGA-NPs in acidic pH (1.2). Prepared with different concentrations of stabilizers, (**A**) PVA and (**B**) PVP.

**Figure 10 pharmaceutics-14-00870-f010:**
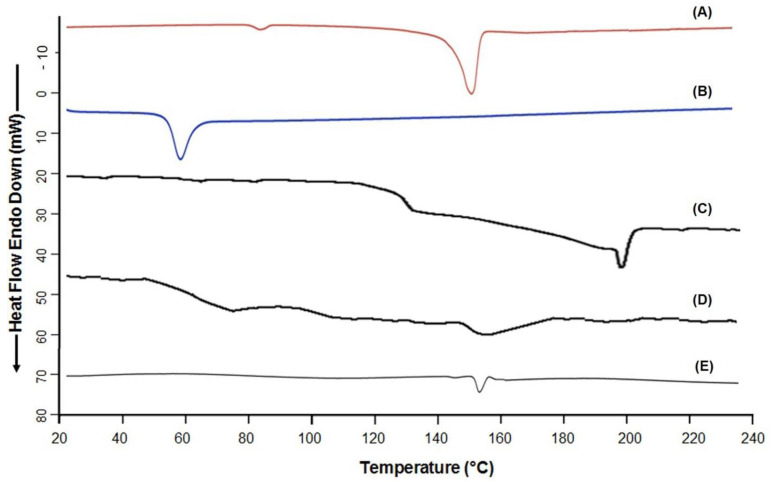
Overlay DSC thermograms of Indomethacin. Indomethacin, IND alone (**A**); PLGA alone (**B**); PVA alone (**C**); PVP alone (**D**) and PLGA-NPs-5 from Batch-2 (**E**).

**Figure 11 pharmaceutics-14-00870-f011:**
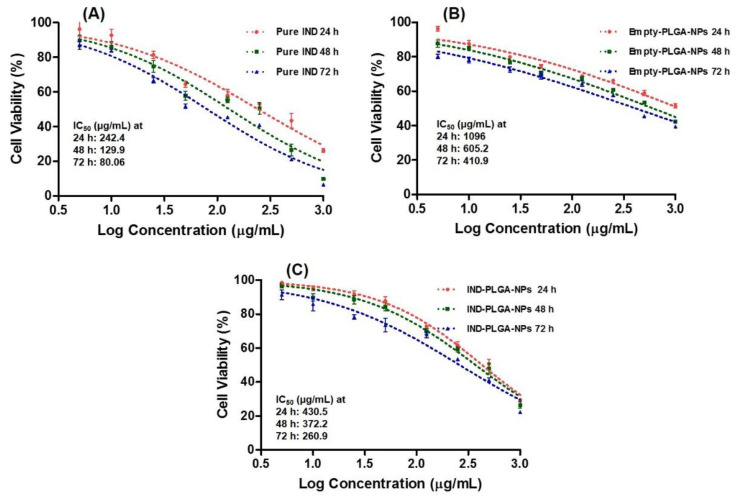
Cytotoxicity after 24 h, 48 h and 72 h of incubation in HepG2 cells: pure IND (**A**); empty-PLGA-NPs (**B**) and IND-PLGA-NPs-5 (**C**) with IC_50_ values (µg/mL).

**Figure 12 pharmaceutics-14-00870-f012:**
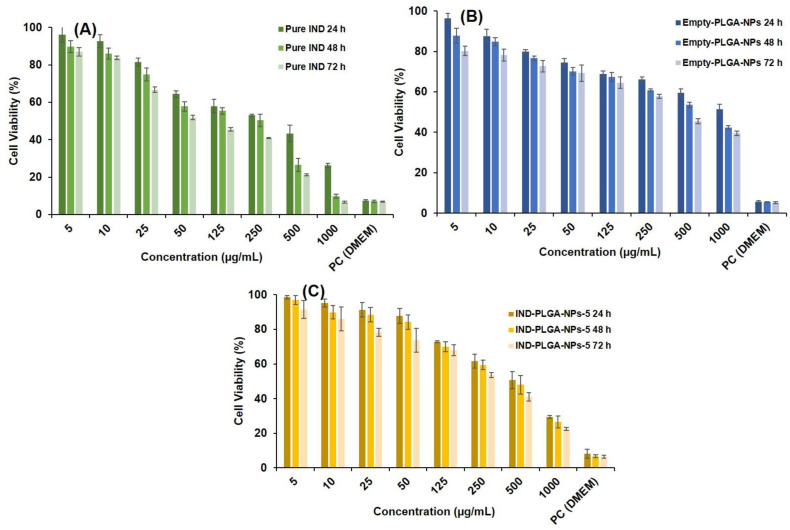
Histograms of the percentage cell viability of HepG2 after 24 h, 48 h and 72 h of incubation with varying concentrations (5–1000 µg/mL) of pure IND (**A**) and its equivalent concentration of empty-PLGA-NPs (**B**) and IND-loaded PLGA-NPs (**C**). Data were presented as the mean of three independent experiments with error bars as standard deviation (SD).

**Table 1 pharmaceutics-14-00870-t001:** PLGA-NPs prepared by the single-emulsion solvent evaporation technique.

Formulations	Organic Solvents *	Type of Stabilizer, Its Concentration *	Mean ± SD *, *n* = 3		
			Particle Size (nm)	Polydispersity Index	Zeta-Potential (mV)
Batch 1					
PLGA-NPs-1	CHCl_3_	PVA (1%, *w*/*v*)	328.1 ± 24.4	0.096 ± 0.079	−1.90 ± 0.91
PLGA-NPs-2	CHCl_3_	PVA (3%, *w*/*v*)	345.2 ± 8.3	0.061 ± 0.036	−1.12 ± 0.06
PLGA-NPs-3	CHCl_3_	PVA (6%, *w*/*v*)	588.1 ± 23.9	0.096 ± 0.028	−0.74 ± 0.12
PLGA-NPs-4	CHCl_3_	PVA (9%, *w*/*v*)	317.6 ± 2.7	0.285 ± 0.088	−0.68 ± 0.11
Batch 2					
PLGA-NPs-5	DCM	PVA (1%, *w*/*v*)	273.2 ± 3.3	0.087 ± 0.064	−0.92 ± 0.21
PLGA-NPs-6	DCM	PVA (3%, *w*/*v*)	406.3 ± 8.5	0.213 ± 0.044	−0.89 ± 0.28
PLGA-NPs-7	DCM	PVA (6%, *w*/*v*)	554.4 ± 23.8	0.239 ± 0.045	−0.76 ± 0.06
PLGA-NPs-8	DCM	PVA (9%, *w*/*v*)	563.9 ± 54.4	0.170 ± 0.015	−0.62 ± 0.58
Batch 3					
PLGA-NPs-9	CHCl_3_	PVP (1%, *w*/*v*)	456.9 ± 61.4	0.152 ± 0.069	17.73 ± 3.45
PLGA-NPs-10	CHCl_3_	PVP (3%, *w*/*v*)	448.8 ± 22.4	0.142 ± 0.035	14.05 ± 2.05
PLGA-NPs-11	CHCl_3_	PVP (6%, *w*/*v*)	466.6 ± 90.5	0.256 ± 0.181	3.86 ± 1.26
PLGA-NPs-12	CHCl_3_	PVP (9%, *w*/*v*)	381.4 ± 83.5	0.255 ± 0.179	2.67 ± 1.41
Batch 4					
PLGA-NPs-13	DCM	PVP (1%, *w*/*v*)	287.8 ± 12.0	0.035 ± 0.029	10.24 ± 3.93
PLGA-NPs-14	DCM	PVP (3%, *w*/*v*)	566.9 ± 64.1	0.268 ± 0.042	5.82 ± 2.65
PLGA-NPs-15	DCM	PVP (6%, *w*/*v*)	609.3 ± 26.9	0.512 ± 0.055	5.37 ± 1.06
PLGA-NPs-16	DCM	PVP (9%, *w*/*v*)	684.2 ± 43.1	0.846 ± 0.016	4.91 ± 2.37

* PVA = Polyvinyl alcohol; PVP = Polyvinyl pyrrolidone; CHCl_3_ = Chloroform; DCM = Dichloromethane; SD = Standard deviation.

**Table 2 pharmaceutics-14-00870-t002:** Physicochemical characteristics of some selected formulations after the encapsulation of Indomethacin (IND). Data are presented as Mean ± SD, *n* = 3.

Formulations	Mean ± SD, *n* = 3								
	Particle Size (nm)	Polydispersity Index	Zeta Potential (mV)	Encapsulation Efficiency (%)	Drug Loading (%)	Cumulative Drug Release (%) at pH 7.4		Cumulative Drug Release (%) at pH 1.2	
						At 12 h	At 72 h	At 12 h	At 72 h
PVA Group formulations									
Batch-1									
1%, PVA with CHCl_3_ (PLGA-NPs-1)	331.8 ± 8.4	0.174 ± 0.035	−2.89 ± 0.95	68.74 ± 5.47	6.24 ± 0.51	16.46 ± 0.63	43.94 ± 2.43	41.28 ± 3.09	42. 16 ± 2.49
3%, PVA with CHCl_3_ (PLGA-NPs-2)	547.4 ± 10.3	0.288 ± 0.078	−0.81 ± 0.14	69.59 ± 6.15	6.32 ± 0.56	13.09 ± 0.78	32.83 ± 1.76	30.38 ± 2.01	32.89 ± 3.88
Batch-2									
1%, PVA with DCM (PLGA-NPs-5)	275.4 ± 8.5	0.157 ± 0.048	−1.13 ± 0.29	72.96 ± 4.59	6.63 ± 0.42	18.82 ± 1.38	52.16 ± 3.83	31.70 ± 2.03	40.06 ± 1.32
3%, PVA with DCM (PLGA-NPs-6)	410.9 ± 11.3	0.205 ± 0.061	−1.14 ± 0.58	74.86 ± 4.78	6.81 ± 0.45	14.58 ± 0.81	37.88 ± 2.11	37.04 ± 1.07	39. 17 ± 1.49
PVP Group formulations									
Batch-3									
1%, PVP with CHCl_3_ (PLGA-NPs-9)	439.3 ± 15.1	0.179 ± 0.033	16.1 ± 3.5	54.94 ± 6.34	4.99 ± 0.58	15.76 ± 0.86	39.90 ± 2.28	34.05 ± 1.27	38.14 ± 1.83
3%, PVP with CHCl_3_ (PLGA-NPs-10)	379.5 ± 10.5	0.177 ± 0.065	15.5 ± 2.6	58.91 ± 4.84	5.36 ± 0.44	16.83 ± 0.95	43.64 ± 2.53	26.46 ± 2.28	32.48 ± 2.12
Batch-4									
1%, PVP with DCM (PLGA-NPs-13)	286.9 ± 13.2	0.066 ± 0.002	10.4 ± 3.94	63.33 ± 6.77	5.76 ± 0.62	19.42 ± 3.47	48.07 ± 2.47	33.13 ± 1.18	35.81 ± 2.08
3%, PVP with DCM (PLGA-NPs-14)	569.6 ± 15.6	0.276 ± 0.055	7.6 ± 2.37	65.37 ± 5.36	5.94 ± 0.49	19.05 ± 1.08	51.68 ± 2.99	39.89 ± 1.86	43.01 ± 1.17

**Table 3 pharmaceutics-14-00870-t003:** Fitting of the in vitro release data in different release kinetic models.

Formulations	Release Kinetic Models														
	Zero Order			First Order			Higuchi Matrix			Korsmeyer–Peppas			Hixon–Crowell		
	*R* ^2^	**k*_0_ (10^−2^)	*n*-Value	*R* ^2^	**k*_1_ (10^−1^)	*n*-Value	*R* ^2^	**k_HM_* (10^−2^)	*n*-Value	*R* ^2^	**k_KP_* (10^−1^)	*n*-Value	*R* ^2^	**k_HC_* (10^−0^)	*n*-Value
PLGA-NPs-1	0.966	1.66	0.0024	0.985	9.31	0.0014	0.998	4.64	0.2366	0.996	1.78	0.2461	0.979	1.38	0.0009
PLGA-NPs-2	0.964	1.28	0.0018	0.979	9.35	0.0009	0.999	3.54	0.0173	0.998	1.35	0.2412	0.974	1.39	0.0006
PLGA-NPs-5	0.968	1.85	0.0029	0.989	9.28	0.0018	0.997	5.26	0.0282	0.994	2.08	0.2563	0.984	1.44	0.0012
PLGA-NPs-6	0.965	1.44	0.0021	0.982	9.33	0.0012	0.999	4.02	0.0203	0.998	1.58	0.2462	0.976	1.42	0.0008
PLGA-NPs-9	0.963	1.43	0.0022	0.982	9.33	0.0013	0.998	4.12	0.0216	0.997	1.65	0.2579	0.976	1.27	0.0009
PLGA-NPs-10	0.964	1.58	0.0024	0.984	9.32	0.0014	0.841	4.51	0.0236	0.998	1.81	0.2562	0.978	1.30	0.0009
PLGA-NPs-13	0.936	1.81	0.0077	0.958	9.29	0.0058	0.991	5.07	0.0822	0.990	2.01	0.0947	0.951	1.34	0.0037
PLGA-NPs-14	0.967	1.76	0.0029	0.959	9.30	0.0022	0.997	5.11	0.0284	0.994	2.02	0.2649	0.983	1.36	0.0012

******k* = Rate constants (*k*_0_ for zero-order, *k*_1_ for first-order, *k_HM_* for Higuchi-Matrix, *k_KP_* for Korsmeyer-Peppas and *K_HC_* for Hixson-Crowell model) and *n*-value = Diffusion exponent.

**Table 4 pharmaceutics-14-00870-t004:** Storage stability of IND-loaded PLGA-NPs (IND-PLGA-NPs-5). Results are represented as mean ± SD of triple measurements.

Characterization Parameters	Storage Time (Days)		
	Initial (0 Day)	At 15th Day	At 30th Day
At 4 °C			
Particle Size (nm)	275.4 ± 8.5	275.8 ± 8.67	276.9 ± 7.81
Polydispersity-index	0.157 ± 0.048	0.158 ± 0.051	0.163 ± 0.054
Zeta-Potential (mV)	−1.13 ± 0.29	−1.12 ± 0.27	−1.11 ± 0.28
Encapsulation efficiency (%)	72.96 ± 4.59	72.54 ± 4.51	71.02 ± 4.63
Drug loading (%)	6.63 ± 0.42	6.50 ± 0.43	6.46 ± 0.45
At 30 °C			
Particle Size (nm)	275.4 ± 8.5	277.2 ± 8.65	279.0 ± 8.81
Polydispersity-index	0.157 ± 0.048	0.158 ± 0.049	0.159 ± 0.047
Zeta-Potential (mV)	−1.13 ± 0.29	−1.12 ± 0.27	−1.11 ± 0.31
Encapsulation efficiency (%)	72.96 ± 4.59	71.87 ± 4.54	71.0 ± 4.64
Drug loading (%)	6.63 ± 0.42	6.53 ± 0.41	6.46 ± 0.42
At 37 °C			
Particle Size (nm)	275.4 ± 8.5	277.8 ± 9.36	279.8 ± 6.95
Polydispersity-index	0.157 ± 0.048	0.159 ± 0.049	0.161 ± 0.048
Zeta-Potential (mV)	−1.13 ± 0.29	−1.12 ± 0.28	−1.11 ± 0.29
Encapsulation efficiency (%)	72.96 ± 4.59	71.38 ± 4.72	70.28 ± 4.22
Drug loading (%)	6.63 ± 0.42	6.44 ± 0.39	6.42 ± 0.41

## Data Availability

The data presented in this study are available on request from the corresponding author.

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
