# Peer review of "Effect of Solvents, Stabilizers and the Concentration of Stabilizers on the Physical Properties of Poly(d,l-lactide-co-glycolide) Nanoparticles: Encapsulation, In Vitro Release of Indomethacin and Cytotoxicity against HepG2-Cell"

_pharmaceutics, 2022, doi:10.3390/pharmaceutics14040870_

Round 1

Reviewer 1 Report

The manuscript 'Effect of solvents, stabilizers and concentration of stabilizers on the physical properties of Poly (D, L-lactide-co-glycolide) nanoparticles: Encapsulation, in vitro release of Indomethacin and cytotoxicity against HepG2-cell' is ready to be accepted by Pharmaceutics.

Author Response

Reviewer # 1

Comments and Suggestions for Authors

The manuscript 'Effect of solvents, stabilizers and concentration of stabilizers on the physical properties of Poly (D, L-lactide-co-glycolide) nanoparticles: Encapsulation, in vitro release of Indomethacin and cytotoxicity against HepG2-cell' is ready to be accepted by Pharmaceutics.

Response: We thank the reviewer for his/her valuable time to read the manuscript thoroughly. After careful reading of the manuscript and giving a bold decision to accept the article.

Reviewer 2 Report

Accept in the current form. 

Author Response

Reviewer # 2

Comments and Suggestions for Authors

Accept in the current form.

Response: We thank the reviewer for his/her valuable time to read the manuscript thoroughly. After careful reading of the manuscript and giving a bold decision to accept the article.

Reviewer 3 Report

The research work entitled Effect of solvents, stabilizers and concentration of stabilizers on the physical properties of Poly (D, L-lactide-co-glycolide) nanoparticles: Encapsulation, in vitro release of Indomethacin and cytotoxicity against HepG2-cell” majorly focuses on the stability/best possible conditions for the PLGA formulation by varying the concentration and types of solvent. In terms of physical significance/properties, the work sounds interesting but as far as the aim and scope of the journal/special issue is concerned, I understand that the submitted manuscript doesn’t match-up to the current level of the journal and its content won’t be able to attract attention of the larger scientific community in pharmaceutical sciences. Therefore, I do not recommend its publication in present form.  

My major suggestions/recommendations to the authors are as follows:

  1. Drug release profile under acidic conditions should be incorporated and compared with pH = 7.4.
  2. In-vitro stability profile of the formulation at 4oC, 37oC and under ambient conditions is not shown/should be incorporated.
  3. HPLC parameters should be incorporated.
  4. In PLGA-NPs-4 (batch 1, table 1), why the particle size is less as compared to 3, 2 and 1? 
  5. FTIR data should be incorporated to support the IND encapsulation.
  6. Figure 9 and 10 are contradictory.
  7. Why PLGA/ or its formulation is showing toxicity?
  8. X- axis of figure 9 is not clearly defined.
  9. IC50 values are not matching with the graph (as per figure 10).
  10. Cytotoxicity assay after 24h and 72h should be incorporated.
  11. Clonogenic assay and scratch assay should be incorporated.
  12. Mechanism of cell death should be shown.
  13. To show extensive studies, 3D cell culture may be incorporated.
  14. Western blot should be incorporated.

Author Response

Reviewer # 3

Comments and Suggestions for Authors

The research work entitled “Effect of solvents, stabilizers and concentration of stabilizers on the physical properties of Poly (D, L-lactide-co-glycolide) nanoparticles: Encapsulation, in vitro release of Indomethacin and cytotoxicity against HepG2-cell” majorly focuses on the stability/best possible conditions for the PLGA formulation by varying the concentration and types of solvent. In terms of physical significance/properties, the work sounds interesting but as far as the aim and scope of the journal/special issue is concerned, I understand that the submitted manuscript doesn’t match-up to the current level of the journal and its content won’t be able to attract attention of the larger scientific community in pharmaceutical sciences. Therefore, I do not recommend its publication in present form.  

We thank the reviewer for his/her valuable comments on our manuscript; the comments and suggestions have greatly improved the quality of the manuscript. We have tried our best to accommodate the recommendations of the learned reviewer in revised manuscript. All the modifications/changes are highlighted as green text in the revised manuscript.

My major suggestions/recommendations to the authors are as follows:

  1. Drug release profile under acidic conditions should be incorporated and compared with pH = 7.4.

Response: Thank you very much for raising this point. The drug release experiment was performed under acidic conditions also. The results obtained were compared with release profile at pH 7.4 and included in the revised manuscript.

  1. In-vitro stability profile of the formulation at 4oC, 37oC and under ambient conditions is not shown/should be incorporated.

Response: The stability study on the optimized formulation was originally performed at three different temperature points viz. 4oC, 30oC and 37oC. We provided the stability profile results at one temperature point, as evidenced in our previously published research such as (REF. 1 & 2 mentioned below). Our lab and other groups have followed the same trend i.e. at one temperature condition (30oC or 37oC) (REF. 3 and 4 mentioned below), also the stability of elastin-PLGA nanoparticle in PBS (at pH 5.5 and 7.4) at 37°C for up to 30 days was done by Stromberg et al., 2021 (REF. 5).

But as suggested by the learned reviewer, we have now incorporated stability profile results at all the three-time points mentioned by the reviewer, and the same has been highlighted in the revised version (Table 4). 

REFERENCES:

REF. 1: The stability of Chitosan-coated poly (lactic-co-glycolide) nanoparticles was performed at 30 ± 1°C. [Abdullah Alshememry, Mohd Abul Kalam, Abdulhadi Almoghrabi, Abdulwahab Alzahrani, Mudassar Shahid, Azmat Ali Khan, Anzarul Haque, Raisuddin Ali, Musaed Alkholief, Ziyad Binkhathlan, Aws Alshamsan. Chitosan-coated poly (lactic-co-glycolide) nanoparticles for dual delivery of doxorubicin and naringin against MCF-7 cells. Journal of Drug Delivery Science and Technology 68 (2022), 103036

REF. 2: Similarly, the stability on Tacrolimus-loaded PLGA-NPs was performed and stored for 30 days at 25±1°C for 30 days only. [Mohd Abul Kalam, Aws Alshamsan. Poly (D, L-lactide-co-glycolide) nanoparticles for sustained release of tacrolimus in rabbit eyes. Biomedicine & Pharmacotherapy 94 (2017) 402–411].

REF. 3: The stability study of atorvastatin calcium loaded PLGA-NPs was evaluated at 37 °C for 10 days. [Z. Li, W. Tao, D. Zhang, C. Wu, B. Song, S. Wang, T. Wang, M. Hu, X. Liu, Y. Wang, The studies of PLGA nanoparticles loading atorvastatin calcium for oral administration in vitro and in vivo, Asian J. Pharm. Sci. 12 (3) (2017) 285–291].

REF. 4: Anita Hafner, Jasmina Lovrić, Dario Voinovich, Jelena Filipović-Grčić. Melatonin-loaded lecithin/chitosan nanoparticles: Physicochemical characterisation and permeability through Caco-2 cell monolayers. Int. J Pharm. 381 (2), 2009; 205-213.

REF. 5: Zachary R. Stromberg, M.Lisa Phipps, Harsha D. Magurudeniy, Christine A. Pedersen, Trideep Rajale, Chris J. Sheehan, Samantha J. Courtney, Steven B. Bradfute, Peter Hraber, Matthew N. Rush, Jessica Z. Kubicek-Sutherland, Jennifer S. Martinez. Formulation of stabilizer-free, nontoxic PLGA and elastin-PLGA nanoparticle delivery systems. Int. J Pharm. 597, 2021, Article # 120340.

  1. HPLC parameters should be incorporated.

Response: The previously reported HPLC-UV method was adopted for the analysis of Indomethacin in the present study (References were included in the manuscript). Now it has been described in brief and included in revised manuscript. The reverse phase (RP) HPLC-UV method was used for the analysis of IND at 241 nm wavelength. “The HPLC system (Waters®, Model-1500-series controller, USA) equipped with UV-detector (Waters®, Model-2489, dual absorbance detector, USA), binary pump (Waters®, Model-1525, USA), automated sampling system (Waters®, Model-2707 Autosampler, USA). The system was run by “Breeze software”. A C18 analytical column (Macherey-Nagel 150 × 4.6 mm, 5 μm) was used at room temperature. The chromatography was performed by isocratic elution of mobile phase composed of 75:25, v/v of Acetonitrile and Milli-Q® water (the pH of water was adjusted to 3.2 by Ortho-Phosphoric acid). The flow rate was 1 mL/min and the volume of injection was 30 μL. Total run time was 5 min and retention time (Rt) for IND was 3.5 min. The calibration curve was constructed in concentration range 0.1, 0.2, 0.4, 0.8, 1.6, 2.5, 5.0 and 10 µgmL-1. The straight line equation obtained was y = 150383x - 6548.7; with a correlation coefficient () of 0.9991.

  1. In PLGA-NPs-4 (batch 1, table 1), why the particle size is less as compared to 3, 2 and 1? 

Response: Thanks for the valuable question raised by the reviewer. We also noticed this phenomenon but we could find a solid reason or exact mechanism, why it could happen. The possible reason what we already discussed previously as “Similarly, an increase in particle size was observed from 317-588 nm as the concentration of PVA was increased and chloroform was used as organic solvent. Contrary to this observation, the particle size was smallest (317 nm) at highest (9%, w/v) concentration of PVA when chloroform was used, which might be due to the strong anchoring of hydrophobic segment of PVA with the matrix of PLGA which remained closely associated with polymer surface [Prabha, S.; Labhasetwar, V. Critical determinants in PLGA/PLA nanoparticle-mediated gene expression. Pharm Res 2004, 21, 354-364].

  1. FTIR data should be incorporated to support the IND encapsulation.

Response: The FTIR spectra of pure drug (IND), PLGA 50:50, empty-NPs and IND-loaded PLGA-NPs-5 were recorded in the range of 4000-450 cm-1 wavenumber. Interpretation of the FTIR scans were done and included in the revised manuscript.

  1. Figure 9 and 10 are contradictory.

Response: Thanks to the reviewer for pointing the remarkable contradiction between these Figures. After performing the MTT assay at 24 h, 48 h and 72 h, the Figure 9 and 10 were plotted (now Figure 11 and 12) again as Logarithm concentration versus % Cell viability. In its present form, results did not indicate any contradiction between the Figures 11 and 12.  

  1. Why PLGA/ or its formulation is showing toxicity?

Response: We have used the concentration of IND 500 µg/mL which was present in 7142.7 µg/mL of PLGA. At this concentration of PLGA, we observed more than 59.47%, 53.48% and 45.55% cells were viable after incubation of 24 h, 48 h and 72 h, respectively. Thus, we can assume that even at this very high concentration of PLGA (empty/ in formulation), the product was non-toxic.

  1. X-axis of figure 9 is not clearly defined.

Response: The Figure 9 is now Figure 11, and the all axes were duly labeled and proper captions have been included.

  1. IC50values are not matching with the graph (as per figure 10).

Response: We have again performed the MTT assay for 24 h, 48 h and 72 h as suggested by the reviewer. Now, all the IC50 values are matching with Figures 11 and 12.

  1. Cytotoxicity assay after 24h and 72h should be incorporated.

Response: The cytotoxicity assay after 24 h, 48 h and 72 h has been incorporated in the manuscript with the revised IC50 values.  

  1. Clonogenic assay and scratch assay should be incorporated.

Response: Our study was designed to synthesize drug-loaded PLGA-NPs using a single-emulsion solvent-evaporation technique using IND as a model lipophilic drug. The physicochemical properties, including the drug encapsulation, it's loading, and in vitro drug release and release kinetics, were investigated in detail. Moreover, the toxicity of PLGA-NPs (with and without IND) was examined by performing the cytotoxicity using HepG2 cells. We understand clonogenic assay and scratch assay are important to study mechanisms of toxicity in vitro, which we hadn't planned for the current study. However, we will consider this for our future studies as we scarcity time and resources..

  1. Mechanism of cell death should be shown.

Response: Evaluating the mechanism of cell death wasn’t the primary objective of our study. As mentioned in the manuscript, we have shown that IC50 of IND alone and IND-PLGANPs were non-toxic to HepG2 cells at therapeutic concentrations, 126.3 µg/mL and 382 µg/mL, respectively. However, IC50 of PLGA-NPs was 698.4 µg/mL. The therapeutic dose below these concentrations was non-toxic and could be used for therapeutic effect. There was no plan to determine the mechanism of cell death as we have used fewer dosages of the developed formulation, drug (IND), and the excipient (PLGA). However, we will consider this suggestion for our future studies.

  1. To show extensive studies, 3D cell culture may be incorporated.

Response: The primary objectives of this study were to synthesize drug-loaded PLGA-NPs using a single-emulsion solvent-evaporation technique using IND as a model lipophilic drug and to perform the cytotoxicity of the IND PLGA and IND-PLGA NPs. We could not find any need to go for 3D cell culture as we are not studying the mechanisms of toxicity induction.

  1. Western blot should be incorporated.

Response:  In the current study, we were not analyzing any protein or gene expressions. Hence, there was no need to perform the Western blot analysis in the present investigation as per our understanding.

Round 2

Reviewer 3 Report

I appreciate authors efforts, but all the suggestions were not properly addressed.

My comments are as follows:

  1. When the free drug is more efficacious as compared to its formulation, then why author's want to encapsulate it inside PLGA, since it is toxic in nature (as per authors explanation and results) but according to the literature, PLGA is not toxic in nature.
  2. According to the authors, PLGA is toxic at some concentration, therefore, they should support their results with its mode of mechanism.
  3. Usually under acidic conditions, the drug release is more as compared to pH = 7.4, but in this study it is reverse. Why? 
  4. Complete 2D cell culture and its mode of mechanism is essential for the complete study of a drug and its formulation. 
  5. Even at low concentration, empty PLGA-NPs is showing some toxicity after 72h (as per figure 12B). Why?

Author Response

Reviewer # 3

Comments and Suggestions for Authors

We thank the reviewer for the valuable comments on our manuscript. The suggestions given by the reviewer have been properly taken care now. The modifications have greatly improved the quality of the manuscript. We have tried our best to respond the comments of the reviewer in revised version of the manuscript. All the changes are highlighted as yellow text in the revised manuscript.

  1. When the free drug is more efficacious as compared to its formulation, then why author's want to encapsulate it inside PLGA, since it is toxic in nature (as per authors explanation and results) but according to the literature, PLGA is not toxic in nature.

Response: The main objective of the present study was to see the effect of solvents, stabilizers, and concentration of stabilizers on the physical properties of PLGA-NPs. Indomethacin as a model lipophilic drug was encapsulated into it just to check the encapsulation efficiency of PLGA-NPs and drug release pattern by considering the first round comments of one of the reviewer. Here, we have performed MTT assay to see if there was any toxic effect of the developed dosage form. At very high concentration, it was found that the PLGA-NPs was little toxic. At very high concentrations/quantity any excipients or drug material can be toxic to any cells. It does not mean that our PLGA-NPs are toxic as perceived by the learned reviewer. The encapsulation of IND in PLGA-NPs reduced the potential toxicity of indomethacin plain drug by 66%, as evident by IC50 of IND alone and IND-PLGANPs were non-toxic to HepG2 cells 126.3 µg/mL and 382 µg/mL, respectively. However, IC50 of PLGA-NPs was 698.4 µg/mL. The therapeutic dose below these concentrations was non-toxic and could be used for the therapeutic effect; these results agree with previous reports.

  1. According to the authors, PLGA is toxic at some concentration, therefore, they should support their results with its mode of mechanism.

Response: Thanks to the reviewer for this comment. At very high concentrations, any excipient or drug material can be toxic. A report indicated that the PLGA-NPs were not significantly toxic at 300 μg/mL against RAW264.7 and BEAS-2B cells (REF 1). In the present investigation, we discussed that “the IND-loaded PLGA-NPs-5, showed increased cell viability at an equivalent drug concentrations (5-1000 µg/mL) and the observed IC50 were 430.5, 372.2 and 260.9 µg/mL at 24 h, 48 h and 72 h, respectively. Our results also indicated similar findings; the difference is that here, we used the term “cell viability.” Also, the cell proliferation assay conducted using empty PLGA-NPs at an equivalent concentration of IND (i.e., amount of empty-PLGA-NPs those containing the same amount of IND in the drug-loaded NPs) has shown IC50 values of 1096, 605.2, and 410.9 µg/mL at 24 h, 48 h, and 72 h, respectively. Here, we explained that even at these high concentrations of PLGA, the cells were viable.

At high concentrations the PLGA-NPs was toxic, this might be associated with the physicochemical properties including the particle size, zeta-potential and selective targeting of the PLGA-NPs. These play an important role in terms of its efficacy in biological systems. In previous reports, it was found that the increasing the particle sizes, the IC50 of cytotoxicity was also increased, which suggests that small sized NPs encouraged higher cytotoxicity at low IC50 values (REF 1-3).

Although, the developed PLGA-NPs in this study did not show particular toxicity at most of the concentrations and could not influence the normal growth of HepG2 cells within the IND dose range (250-500 µg/mL) and at the equivalent amount of PLGA-NPs (3571.4-7142.7 µg/mL). Conclusively, an ideal size, zeta-potential, dose, duration of treatment, and targeting ability of PLGA-NPs to deliver the encapsulated drugs at the desired site are essential to developing efficient and safe drug carriers.

[REF 1: Sijing Xiong, Saji George, Haiyang Yu, Robert Damoiseaux, Bryan France, Kee Woei Ng, Joachim Say-Chye Loo. Size influences the cytotoxicity of poly (lactic-co-glycolic acid) (PLGA) and titanium dioxide (TiO2) nanoparticles. Arch Toxicol. 2013 Jun; 87(6): 1075–1086].

[REF 2: Hock Ing Chiu, Nozlena Abdul Samad, Lizhen Fang, Vuanghao Lim. Cytotoxicity of targeted PLGA nanoparticles: a systematic review. RSC Adv., 2021, 11, 9433. DOI: 10.1039/d1ra00074h].

[REF 3:  Di-Wen S, Pan GZ, Hao L, Zhang J, Xue QZ, Wang P, Yuan QZ. Improved antitumor activity of epirubicin-loaded CXCR4-targeted polymeric nanoparticles in liver cancers. Int. J Pharm. 2015, 500 (1-2): 54-61. DOI: 10.1016/j.ijpharm.2015.12.066 PMID: 26748365].

  1. Usually under acidic conditions, the drug release is more as compared to pH = 7.4, but in this study it is reverse. Why? 

Response: Thank you very much for raising this point. The release of drug at 12 h, was very high at acidic pH as compared to pH 7.4 as shown in Table 2. The release of drug at acidic and basic conditions are almost equal from PLGA-NPs-1, PLGA-NPs-2 and PLGA-NPs-6 at 72 h study. While from other formulations the drug release was comparatively higher at pH 7.4 at 72 h, which could be attributed to high drug release during the initial hours at acidic condition, might saturated the release even after maintaining the sink condition. Thus, further release from the formulation could not occurred, so the release in acidic condition was low in some formulations. Moreover, the degradation mechanism of PLGA at pH 7.4 takes place “inside-out” and at acidic pH it occurs as "outside-in". At pH 7.4, in some PLGA-NPs the pore formation and surface pitting might occurred. Whereas, at acidic pH the PLGA-NPs maintained their smooth surfaces during the process of polymer degradation and rupturing. The rupturing of the PLGA-NPs was accredited to the crystallization of the oligomeric degradation of PLGA due to its low solubility at acidic pH. Also, Zolnik and Burgess, 2007, reported that the degradation of PLGA taken place in an extra homogeneous pattern at acidic pH as compared to pH 7.4. This might be the result of entire NPs experienced a close-to-uniform pH at acidic condition. However, at pH 7.4, the local micro environmental pH within the PLGA-NPs was varied significantly due to accumulation of acid oligomers. Thus, a heterogeneous degradation resulted in the random creation of pores within PLGA-NPs degraded at pH 7.4 which was not occurred during their degradation acidic pH.

[REF: Banu S. Zolnik, Diane J. Burgess. Effect of acidic pH on PLGA microsphere degradation and release. Journal of Controlled Release. 122 (3), 2007, 338-344. (PMID: 17644208)].

  1. Complete 2D cell culture and its mode of mechanism is essential for the complete study of a drug and its formulation. 

Response: The culture of HepG2 cell was maintained to perform the MTT assay to check the potential toxicity (if any) of indomethacin (IND) and PLGA-NPs (with and without indomethacin).  At very high concentration, any excipient or drug material can be toxic. A report indicated that the PLGA-NPs was not significantly toxic at 300 μg/mL against RAW264.7 and BEAS-2B cells (REF 1).

In the present investigation, we discussed that “the IND-loaded PLGA-NPs-5, showed increased cell viability at equivalent drug concentrations (5-1000 µg/mL) and the observed IC50 were 430.5, 372.2 and 260.9 µg/mL at 24 h, 48 h and 72 h, respectively. Our results also indicated the similar findings the cell proliferation assay conducted using empty-PLGA-NPs at equivalent concentration of IND (i.e. amount of empty-PLGA-NPs those containing the same amount of IND in the drug-loaded NPs) has shown IC50 values of 1096, 605.2 and 410.9 µg/mL at 24 h, 48 h and 72 h, respectively. Here, we explained that even at these high concentrations of PLGA, the cells were viable. Although at very high concentrations, the PLGA-NPs was little toxic, this might be associated with the physicochemical properties including the particle size, zeta-potential and selective targeting of the PLGA-NPs. These also play an important role in terms of PLGA-NPs efficacy in biological systems. Although, this study was designed to synthesize drug-loaded PLGA-NPs using a single-emulsion solvent-evaporation technique using IND as a model lipophilic drug, to see the effect of solvents, stabilizers and concentration of stabilizers on the physical properties of PLGA-NPs. Therefore, particle characterization and morphology were observed in detail. As per the reviewer’s suggestion, a model lipophilic drug (indomethacin) was encapsulated into the PLGA-NPs and in vitro drug release and release kinetics of the drug were performed in detail.

Also, the cytotoxicity potential of PLGA-NPs (if any) was checked against HepG2-cell. HepG2 is a liver-derived human cell line that displays abundant parenchymal cell functions containing the metabolism of polycyclic aromatic hydrocarbons. The observed proliferation rate of HepG2 (i.e., doubling time) was measured, and it was found that these cells double after 48.95 h at a cell density of HepG2 cells (5×105 cells/well plate). The highest live cells were observed at 72 h, and these results were similar to Miura et al., 1999, who has seeded 4×105 cells/well in 6-well plates, and they believed that this was a more appropriate cell number for 6-well plates (REF 2). The doubling time of HepG2 cells was 48.95 h in the present investigation, which was consistent with other studies, where the doubling time of HepG2 was 2.05 days (49.2 h) (REF 3-5). The growth rate at 24 h, 48 h, and 72 h were 53%, 52%, and 49%, respectively, and there was no major change in the growth rate with cell viability of more than 94.27% at 72 h. Thereby, no significant change in doubling time was found.

[REF 1: Sijing Xiong, Saji George, Haiyang Yu, Robert Damoiseaux, Bryan France, Kee Woei Ng, Joachim Say-Chye Loo. Size influences the cytotoxicity of poly (lactic-co-glycolic acid) (PLGA) and titanium dioxide (TiO2) nanoparticles. Arch Toxicol. 2013 Jun; 87(6): 1075–1086].

[REF 2: N Miura, Y Matsumoto, S Miyairi, S Nishiyama, A Naganuma. Protective effects of triterpene compounds against the cytotoxicity of cadmium in HepG2 cells. Mol Pharmacol. 1999, 56 (6): 1324-1328. doi: 10.1124/mol.56.6.1324].

[REF 3: Melva Louisa, Frans D Suyatna, Septelia Inawati Wanandi, Puji Budi Setia Asih, Din Syafruddin. Differential expression of several drug transporter genes in HepG2 and Huh-7 cell lines. Adv Biomed Res. 2016, 5:104. doi: 10.4103/2277-9175.183664. eCollection 2016. DOI: 10.4103/2277-9175.183664].

[REF 4: Norouzzadeh M, Kalikias Y, Mohammadpour Z, Sharifi L, Mahmoudi M. Determining population doubling time and the appropriate number of HepG2 cells for culturing in 6- well plate. Int. Res. J Applied Basic Sci. 2016, 10: 299-303].

[REF 5: Wen-Sheng W. ERK signaling pathway is involved in p15INK4b/p16INK4a expression and HepG2 growth inhibition triggered by TPA and Saikosaponin a. Oncogene. 2003;22(7):955-963].

  1. Even at low concentration, empty PLGA-NPs is showing some toxicity after 72h (as per figure 12B). Why?

Response:  In this study, the empty PLGA-NPs did not show any particular toxicity even after 72 h, as represented in Figure 12B. Thanks for the reviewer’s suggestion. As per our observation, there was no toxicity as evident from IC50 of indomethacin alone and indomethacin loaded PLGA-NPs (126.3 µg/mL and 382 µg/mL, respectively). So, these were non-toxic to HepG2 cells. This means the toxicity of IND was reduced by 66% in case of formulation (IND-PLGA-NPs). We have used the concentration of IND (500 µg/mL) which was present in 7142.7 µg/mL of PLGA. At this concentration of PLGA, we observed more than 59.47%, 53.48% and 45.55% cells were viable after incubation of 24 h, 48 h and 72 h, respectively during the MTT assay. Thus, we can assume that even at this very high concentration of PLGA (in empty PLGA-NPs/ IND-PLGA-NPs-5) the product was non-toxic.

This manuscript is a resubmission of an earlier submission. The following is a list of the peer review reports and author responses from that submission.

Round 1

Reviewer 1 Report

The Authors described the effect of different conditions in the production of PLGA nanoparticles. In particular, Alkholief and colleague compared some characteristics of NP-PLGA obtained using two different solvents and stabilizers during the NP production step.

After a careful consideration, the reviewer regrets to inform that, in his opinion, the manuscript is not appropriate for this journal. The main reason is the lack of novelty since the characteristics and the procedures to obtain NP-PLGA have been extensively described in the literature and the study conducted by the authors has a very limited significance.

Secondly, the manuscript is confusing in some parts and the results’ description is unclear and not supported by a proper statistical analysis.

In details:

  • Paragraph 3.1: It is just a literature review, no results from the Authors have been described.
  • Line 196-200: What is the method used for the particle size analysis? Has it been done by DLS? If so, why did the subsequent analysis shown in table 2 was made through the binning method?
  • Line 208-210: It is not clear what the Authors tried to say.
  • Table 1: What is the reason why in batch 2 a higher percentage of stabilizer corresponds to a higher particle size distribution? There is no discussion regarding this finding in the paper.
  • Table 2: Looking at the SD and the average size, it seems that there is not a significative difference among the particle characteristics. Did the Author perform a proper statistical analysis?
  • Line 218-202: The Authors stated that “PLGA-NPs prepared by using PVA as stabilizer are smaller as compared to PVP….” But the results showed in table 1 do not match with this sentence.
  • Line 228-235: In the opinion of the reviewer, the explanation given to explain the results found on batch 1 is not clear and does not describe the findings properly.
  • Line 243-246: It is hard to understand what the Authors tried to say.
  • Line 270-272: What is the correlation between the aggregate formation and the water phase viscosity in this case?
  • Line 282-284: The Authors stated “The presence of PVA at the interface prevented the coalescence of nano-droplets and aggregation of NPs, which might be due to reduction of the overall energy of the two phases by PVA”. Why was it not true for the PVP?
  • Line 322-325: The authors stated that the viscosity of aqueous solutions of PVA and PVP increased with increasing their concentrations but the solutions containing 1% of polymer are more viscous than the ones containing 3% as showed in figure 6.
  • Line 346-347: What is the reason of the presence of the peak at 250C?
  • Line 351-352: “Furthermore, non-appearance of additional peaks PVA and PVP indicating that the stabilizers were found to be compatible with PLGA.”. What did the Authors mean whit this sentence? In which way do extra peaks represent an incompatibility between the materials?
  • Other comments: The manuscript presents a number of typos.

Reviewer 2 Report

The manuscript 'Effect of solvents, stabilizers and concentration of stabilizers on the physical properties of Poly (D, L-lactide-co-glycolide) nanoparticles' presented a preparation method of nano-particle for readers. The solvent, stabilizer and solvent concentration effects were optimized. 

The purpose of nano-particle preparation is to controllably load and release drugs. The method herein didn't have a model drug which limits the meaning of this manuscript. 

  1. The diameters of nano-particles in Table 2 and Table 1 have obvious differences with similar preparation. However, the differences weren't discussed in detail. 
  2. The dissolution release test is an important indicator for nanoparticles, and it is difficult to evaluate the pros and cons of nanoparticles in the current data.

Reviewer 3 Report

The manuscript titled 'Effect of solvents, stabilizers and concentration of stabilizers on the physical properties of Poly (D, L-lactide-co-glycolide) nanoparticles' contains some valuable information but there was a similar study published by Hernández-Giottonini et all (DOI: 10.1039/C9RA10857B (Paper) RSC Adv., 2020, 10, 4218-4231) in 2020 in RSC advances that authors may want to look at.  

Manuscript is also a bit low on data for example data such as zeta potential, PDI etc can be obtained very easily. data could be presented differently i.e. use of graphs etc to provide information on the correlation between concentrations etc.  

I also think that this manuscript needs thorough proofreading. Few points I picked up; use of ‘by using’ – in most places you can convey the message with either ‘by’ or ‘using’, you do not need both. Also, section 3.1 has at least 20 ‘used’ in the paragraph, please rewrite.

Few other points authors may want to work with and/or clarify:

Material and method:

Line 92: What does power 4 mean? amplitude and was it straight 40 seconds of sonicaiton or you did this in cycles? 

Line 106: Write a brief method even if you are using a well-established method from somewhere else.

Line 135: The hermetic seals were placed- replace hermetic seals with hermetically sealed pans.

Results and discussion:

Section 3.1: This section reads like a literature survey and isn't directly related to any results. Although useful information is presented here but it is placed randomly as a section without any relevance to what you have done. This could make a part of the discussion for sections later on and doesn’t need to be presented as one separate section.

Delete sentence ‘PLGA used….’ Lines 146-147.

Lines 197,198,200,201: Avoid decimal points. No method/technique is this accurate and you are providing a range here anyway.

Line 209-210: You may want to elaborate the huge difference between the PS with two different analysis, e.g. limitation of binning analysis.

Line 234-235: Should you present Zeta potential values of your particles? It will be easier to explain/justify this.

Line 241-243: Correct the statement.

Line 250: closed bracket?

Line 287: particles do not appear to be unimodal by looking these micrographs. Do you have PDI data you could use to justify this?

Line 349: The PVP/PVA content may be well below the detection limit! Also, why some DSCs are run till 250 but others till 280? Clarify this in the method

Figure 7: looks a bit messy, plot it yourself on Excel or Origin.